# Structure-Aware *Mycobacterium tuberculosis* Functional Annotation Uncloaks Resistance, Metabolic, and Virulence Genes

Samuel J. Modlin,[a] Afif Elghraoui,[a] Deepika Gunasekaran,[a] Alyssa M. Zlotnicki,[a] Nicholas A. Dillon,[b] Nermeeta Dhillon,[a] Norman Kuo,[a] Cassidy Robinhold,[a] Carmela K. Chan,[a] Anthony D. Baughn,[b] Faramarz Valafar[a]

[a]Laboratory for Pathogenesis of Clinical Drug Resistance and Persistence, San Diego State University, San Diego, California, USA
[b]Department of Microbiology and Immunology, University of Minnesota Medical School, Minneapolis, Minnesota, USA

**ABSTRACT** Accurate and timely functional genome annotation is essential for translating basic pathogen research into clinically impactful advances. Here, through literature curation and structure-function inference, we systematically update the functional genome annotation of *Mycobacterium tuberculosis* virulent type strain H37Rv. First, we systematically curated annotations for 589 genes from 662 publications, including 282 gene products absent from leading databases. Second, we modeled 1,711 underannotated proteins and developed a semiautomated pipeline that captured shared function between 400 protein models and structural matches of known function on Protein Data Bank, including drug efflux proteins, metabolic enzymes, and virulence factors. In aggregate, these structure- and literature-derived annotations update 940/1,725 underannotated H37Rv genes and generate hundreds of functional hypotheses. Retrospectively applying the annotation to a recent whole-genome transposon mutant screen provided missing function for 48% (13/27) of underannotated genes altering antibiotic efficacy and 33% (23/69) required for persistence during mouse tuberculosis (TB) infection. Prospective application of the protein models enabled us to functionally interpret novel laboratory generated pyrazinamide (PZA)-resistant mutants of unknown function, which implicated the emerging coenzyme A depletion model of PZA action in the mutants' PZA resistance. Our findings demonstrate the functional insight gained by integrating structural modeling and systematic literature curation, even for widely studied microorganisms. Functional annotations and protein structure models are available at https://tuberculosis.sdsu.edu/H37Rv in human- and machine-readable formats.

**IMPORTANCE** *Mycobacterium tuberculosis*, the primary causative agent of tuberculosis, kills more humans than any other infectious bacterium. Yet 40% of its genome is functionally uncharacterized, leaving much about the genetic basis of its resistance to antibiotics, capacity to withstand host immunity, and basic metabolism yet undiscovered. Irregular literature curation for functional annotation contributes to this gap. We systematically curated functions from literature and structural similarity for over half of poorly characterized genes, expanding the functionally annotated *Mycobacterium tuberculosis* proteome. Applying this updated annotation to recent *in vivo* functional screens added functional information to dozens of clinically pertinent proteins described as having unknown function. Integrating the annotations with a prospective functional screen identified new mutants resistant to a first-line TB drug, supporting an emerging hypothesis for its mode of action. These improvements in functional interpretation of clinically informative studies underscore the translational value of this functional knowledge. Structure-derived annotations identify hundreds of high-confidence candidates for mechanisms of antibiotic resistance, virulence factors, and basic metabolism and other functions key in clinical and basic tuberculosis research.

Address correspondence to Faramarz Valafar, faramarz@sdsu.edu.

More broadly, they provide a systematic framework for improving prokaryotic reference annotations.

**KEYWORDS** *Mycobacterium tuberculosis*, annotation, structure, virulence factors, functional genomics, pyrazinamide, resistance, antibiotic resistance, protein structure-function

Manual curation remains the gold standard for annotating function from literature (1), yet requires massive effort from highly specialized researchers. UniProt curators alone evaluate over 4,500 papers each year (1). Literature annotation is typically complemented with functional inference by sequence homology, but this approach fails to identify distant relatives (remote homologs) or convergently evolved proteins of shared function (structural analogs).

These challenges hinder the study of *Mycobacterium tuberculosis*, the etiological agent of tuberculosis (TB). The *M. tuberculosis* virulent type strain H37Rv, a descendant of strain H37, was isolated from a pulmonary TB patient in 1905 and kept viable through repeated subculturing (2). Following sequencing of the H37Rv genome, function was assigned to 40% of its open reading frames (ORFs) (3) and then expanded to 52% in 2002 following reannotation (4). New annotations continued to be added by TubercuList (now part of Mycobrowser, https://mycobrowser.epfl.ch/) until March 2013. To date, one-quarter of the H37Rv genome (1,057 genes) lacks annotation entirely, listed in "conserved hypotheticals" or "unknown" functional categories, and hundreds more annotations minimally describe product function (e.g., "possible membrane protein"). Though other databases have emerged in recent years (5–9), Mycobrowser remains the primary resource for gene annotation for TB researchers (10) yet lacks functional characterizations reported in the literature.

Moreover, many proteins key to *M. tuberculosis* pathogenesis are challenging to ascribe function to by sequence similarity. For instance, transport proteins—many of which allow *M. tuberculosis* to tolerate drug exposure by effluxing drug out of the cell (11)—have membrane-embedded regions under relaxed constraint compared to globular proteins and diverge in sequence more rapidly as a result (12). This rapid divergence challenges their characterization through homology. Limitations of sequence-based approaches to detect and annotate *M. tuberculosis* proteins motivate an alternative approach to annotating *M. tuberculosis* gene function.

One alternative approach is identifying functional protein homologs and analogs through shared structure, which offers considerable advantages. This approach mitigates bias toward *a priori* assumptions by not limiting search space to evolutionarily close relatives, enabling discovery of functions shared between structurally similar proteins of distant homology, or analogy between protein structures without a common ancestor. This can be especially valuable for inferring function at the host-pathogen interface, which is challenging to recapitulate in the laboratory. Moreover, analogous or distantly homologous relationships between proteins of shared structure/function are challenging to resolve by sequence similarity, as they evolve convergently or, in the case of distant homology, have significant changes in sequence over long periods of evolution, resulting in shared structure and function despite low amino acid (AA) similarity (13).

Iterative Threading ASSEmbly Refinement (I-TASSER) (14) builds three-dimensional protein structure from sequence through multiple threading alignment of the Protein Data Bank (PDB) (15) templates, followed by iterative fragment assembly simulations. I-TASSER accurately predicts structure (16–20), provides metrics for model quality (21) (C-score) and pairwise structural similarity (22) (TM-score), and integrates function and structure prediction tools (23) (COACH and COFACTOR) comprising Gene Ontology (GO) terms (24), Enzyme Commission (EC) numbers (25), and ligand binding sites (LBS) (26).

EC numbers and GO terms partially or completely define gene function and are widely incorporated into mainstream databases. EC numbers describe catalytic function hierarchically through a four-tiered numerical identifier system that funnels from general enzyme class (e.g.,

**TABLE 1** Comparison among frequented annotation resources[a]

| Metric | TubercuList | PATRIC | RefSeq | Mtb Network Portal | UniProt | KEGG | BioCyc |
|---|---|---|---|---|---|---|---|
| Coding sequences | 4,038 | 4,367 | 3,989 | 4,038 | 3,997 | 3,906 | 4,031 |
| Proteins with functional assignments | 2,815 | 3,007 | 2,341 | 2,853 | 2,906 | 1,750 | 2,571 |
| Hypothetical proteins | 1,223 | 1,360 | 1,648 | 1,185 | 1,091 | 2,156 | 1,460 |
| Proteins with ≥1 GO term | 2,629 | 969 | 0 | 2,460 | 3,305 | 0 | 3,557 |
| Proteins with EC no.(s) assigned | 1,293 | 1,074 | 1,081 | 1,003 | 1,138 | 1,050 | 1,018 |

[a]"Functional assignments" refer to annotations that describe protein function and exclude hypothetical, unknown/uncharacterized, and PE/PPE family proteins. Counts reflect database content on 17 May 2017 for RefSeq (36) (https://www.ncbi.nlm.nih.gov/refseq/), PATRIC (6) (https://www.patricbrc.org/), and Mtb Network Portal (9) (http://networks.systemsbiology.net/mtb/) and 23 June 2017 for KEGG (120) (https://www.kegg.jp/kegg/genome/pathogen.html) and UniProt (116) (https://www.uniprot.org/uniprot/). The number of CDS in KEGG is reported as 3,906 because they include only protein-coding genes. The source of annotations for *M. tuberculosis* protein-coding genes in KEGG is TubercuList (131).

ligase, oxidoreductase) down to substrate specificity with atomic precision (25). GO terms add to EC number content: they describe gene products by where they function, the processes they are involved in, and their specific molecular function in species-independent form (27, 28). This cross-species unification is particularly useful for reconciling annotation transfers of analogs and distant homologs into gene product names.

Previous hypothetical gene annotation efforts for *M. tuberculosis* have not included a systematic manual literature curation component and have drawn from inferential techniques such as protein homology and fold similarity (29, 30), aggregating gene orthology server predictions (31), metabolic pathway gap-filling (32), and STRING interactions (33), lacking inclusion criteria based on benchmarked likelihood of correctness. Measured interpretation of annotated gene functions requires the source of the annotation and the reliability of the evidence warranting it to be described explicitly. We strived to provide this resource by reconciling the H37Rv annotation on Mycobrowser with published functional characterization and systematically inferring function from structural similarity to annotate genes challenging to characterize through experiment and sequence analysis. We include orthogonal validation measures to confidently capture unexpected functions while minimizing "overannotation" (34, 35).

We report our findings in three sections. First, we establish the set of underannotated genes, describe our systematic manual literature curation protocol, and summarize the novelty of the resulting annotations with respect to popular functional databases. Next, we describe our structural modeling pipeline, orthogonal validation and quality assurance methods, and two illustrative examples of manually curated functional annotations from structural inferences unsupported by an established method of detecting remote functional homology. Finally, we summarize the updated annotation and genes remaining to be characterized and demonstrate the added value of this annotation through its application to previously published and novel functional screens.

## RESULTS

**Numerous genes lack annotation in all common *M. tuberculosis* databases.** First, we defined a set of 1,725 underannotated genes (see Data Set S1 in the supplemental material) based on their TubercuList entry. We included

1. Genes in "conserved hypothetical" or "unknown" functional categories.
2. Genes qualified by an adjective connoting low confidence (e.g., "predicted" or "possible").
3. Genes described by something other than function (e.g., "alanine-rich protein" or "isoniazid-inducible protein").
4. Genes of the PE/PPE family—a largely uncharacterized, polymorphic protein family unique to mycobacteria with proline-glutamine or proline-proline-glutamine N-terminal domains.

Next, we asked how many of these genes lacked annotations across commonly referenced databases (Table 1). Although BioCyc and UniProt had more genes with GO terms than TubercuList, and UniProt and Mtb Network Portal had fewer hypothetical

proteins than TubercuList, all databases had over one-quarter of coding sequences (CDS) annotated as hypothetical, demonstrating the need for systematic manual annotation.

**Frequently consulted annotation sources lack experimentally demonstrated functions.** We devised a manual curation protocol (Text S1 and Fig. S1) that

1. Assigns qualifying adjectives that connote confidence.
2. Assigns Enzyme Commission ("EC") numbers.
3. Requires multiple reviewers per paper to hedge against human error, and an additional quality control curator to check formatting and annotation consistency.

We systematically reviewed over 5,000 publications according to this protocol, furnishing annotations for one-third of underannotated genes (575) with product function or functional notes (Data Set S1). Of these, 282 were annotated with product function absent from TubercuList, including 122 enzymes and 28 regulatory proteins. These annotations include 14 oxidative stress response genes, 22 proteins mediating RNA and DNA functions, and eight transport/efflux proteins.

Next, we evaluated whether these missing annotations were restricted to TubercuList or more widespread. We checked our curations against four frequented annotation resources: UniProt (Data Set S2), Mtb Network Portal (9), PATRIC (6), and RefSeq (36) (Data Set S2). Product function information was absent from 172/282 (61%) of these genes on UniProt (Data Set S2), and 118 (64 of which are antigens [Text S1 and Fig. S4A]) were more thoroughly annotated than in any of the examined databases (Data Set S2). This novelty underscores the value of these manual curations and highlights critical information that these databases lack (Table 2 and Data Set S1). After excluding antigens, 25.2% of genes with function curated from literature were absent from all five annotation resources. To identify enzymatic functions unannotated elsewhere, we compared our manual EC number assignments to commonly referenced databases (Text S1 and Fig. S4). This comparison revealed that 59/98 (60.2%) of genes assigned EC numbers have EC numbers only in our annotation. These missing annotations include functions affecting drug resistance, features of *in vivo* infection, and other important functions. Examples include a rare instance where a PE/PPE gene has demonstrated catalytic function (37) (Rv1430), a probable peptidoglycan hydrolase implicated in isoniazid (INH) and pyrazinamide (PZA) resistance and biofilm formation (38) (Rv0024), a rhomboid protease with roles in biofilm formation and ciprofloxacin and novobiocin resistance (39) (Rv1337), and an oxidoreductase important for in-host survival of *M. tuberculosis* (40–42) (Rv3005c). Additional findings pertinent to pathogenesis, host-pathogen interaction, and antibiotic resistance were noted across underannotated genes (Data Set S4).

**Annotating function from structure similarity.** Next, we modeled protein structures and developed a procedure to annotate function based on shared structure according to the likelihood that two proteins shared function (i.e., precision [equation 1, Materials and Methods]). To inform our annotation methods we first assessed whether we could

1. Reliably infer precision according to similarity.
2. Differentiate between precision thresholds at different levels of functional detail (e.g., EC number tiers).

To make these assessments, we benchmarked precision as a function of template-modeling score (TM-score), a measure of structural similarity independent of protein length, and sequence similarity (amino acid identity [AA%]), using a set of 363 genes with known function (Materials and Methods) through the standalone version of I-TASSER. TM-score and AA% were predictive of precision and mutually correlated ($R = 0.784$, Pearson correlation coefficient) among both concordant and discordant EC numbers (Text S1 and Fig. S2B). We accounted for TM-score and AA% simultaneously by their geometric mean ($\mu_{geom}$) to estimate precision. Precision of EC number prediction increased monotonically as a function of $\mu_{geom}$ for all 4 EC tiers, and regression lines for the 4 degrees of EC functional specificity did not intersect (Fig. 1). From these properties we concluded that we could reliably estimate precision from $\mu_{geom}$ with distinct thresholds for each EC tier. We defined thresholds as the $\mu_{geom}$ value where logistic regression lines intersected with 50% for receiving "putative" and 75% for receiving

**TABLE 2** Genes annotated through systematic manual curation that expand upon annotations from major databases[a]

| Gene | Product | PMID(s) | TubercuList | UniProt | Mtb Network Portal |
|---|---|---|---|---|---|
| **Novel** | | | | | |
| Rv0309 | Adhesin/putative L,D-transpeptidase | 23922800, 23889607, 26201501 | Possible conserved exported protein | Possible conserved exported protein | |
| Rv0394c | Hyaluronidase/chondrosulfatase | 23465892 | Possible secreted protein | Possible secreted protein | Possible membrane protein |
| Rv0431 | Probable vesiculogenesis/immune response regulator | 24248369, 21170273, 17436267, 26324094, 27765619 | Putative tuberculin-related peptide | Putative tuberculin related peptide | Tuberculin-related peptide |
| Rv1430* | Esterase | 23383323 | PE family protein PE16 | PE family protein PE16 | PE family protein |
| Rv1993c | Putative chaperone | 21925112 | Conserved protein | Uncharacterized protein | |
| Rv2345 | Probable phosphatase | 25782739 | Possible conserved transmembrane protein | UPF0603 protein | Possible membrane protein |
| Rv2923c | Probable osmotically induced bacterial protein C (OsmC, a probable peroxide reductase) | 22088319 | Conserved protein | Uncharacterized protein | |
| Rv2954c | Probable methyltransferase | 23536839 | Hypothetical protein | Uncharacterized protein | |
| Rv2969c | Periplasmic disulfide-bond-forming (Dsb) enzyme | 24100317, 18539140 | Possible conserved membrane or secreted protein | Membrane protein (possible conserved membrane or secreted protein) | Possible conserved membrane or secreted protein |
| Rv3528c | Probable serine hydrolase | 26853625 | Unknown protein | Uncharacterized protein | |
| **Greater specificity** | | | | | |
| Rv0059 | Probable toxin DarT/probable DNA ADP-ribosyltransferase | 27939941 | Hypothetical protein | Uncharacterized protein | |
| Rv0060 | Probable antitoxin DarG/probable DNA ADP-ribosylglycohydrolase | 27939941 | Conserved hypothetical protein | Uncharacterized protein | ADP-ribose 1-phosphate phosphatase-related protein |
| Rv1337 | Probable rhomboid protease/integral membrane protein | 19165721, 23029216 | Probable integral membrane protein | Uncharacterized protein | Rhomboid membrane family protein |
| Rv1357c | Cyclic diguanylate phosphodiesterase | 21151497 | Conserved hypothetical protein | Uncharacterized protein | Sensory box/GGDEF family protein |
| Rv1566c | Probable noncatalytic peptidoglycan binding RipD protein/probable antigen | 24107184, 26481294 | Possible Inv protein | Possible Inv protein | Invasion protein |
| Rv2024c | Restriction enzyme/m-6-adenine DNA methyltransferase (mycobacterial adenine methyltransferase B "MamB") | 26704977 | Conserved hypothetical protein | Uncharacterized protein | Putative helicase |
| Rv2695 | Probable serine hydrolase | 26853625 | Conserved hypothetical alanine-rich protein | Conserved hypothetical alanine-rich protein | |
| Rv2991 | Probable flavin/deazaflavin oxidoreductase | 26434506 | Conserved protein | Conserved protein (F420-dependent protein) | |

**TABLE 2** (Continued)

| Gene | Product | PMID(s) | TubercuList | UniProt | Mtb Network Portal |
|---|---|---|---|---|---|
| Rv3036c | Esterase | 25224799 | Probable conserved secreted protein TB22.2 | Probable conserved secreted protein TB22.2 | Possible membrane protein |
| Rv3354 | Protein kinase | 25139900 | Conserved hypothetical protein | Lipoprotein | Possible lipoprotein LprJ |
| Orthogonal annotation | | | | | |
| Rv0256c* | B cell antigen/probable INOS promoter binding protein | 23827809, 28071726 | PPE family protein PPE2 | Uncharacterized PPE family protein PPE2 | Predicted cobalt transporter in mycobacteria |
| Rv2204c | Probable serine hydrolase | 26298037, 26853625, 26536359 | Conserved protein | Protein Rv2204c | Probable iron binding protein from the HesB_IscA_SufA family |
| Rv3779 | Polyprenylphosphomannosyl synthase/galactosaminyltransferase | 21030587, 19717608 | Probable conserved transmembrane protein, alanine and leucine rich | Membrane protein (probable conserved transmembrane protein, alanine and leucine rich) | |

[a]Annotations are separated into completely novel, those with similar annotations but with greater specificity, and those with an additional, orthogonal annotation compared to evaluated databases (Text S1 and Fig. S4A). PubMed IDs (PMIDs) from which annotations for each product were derived are included. Members of the PE/PPE family are indicated by asterisks.

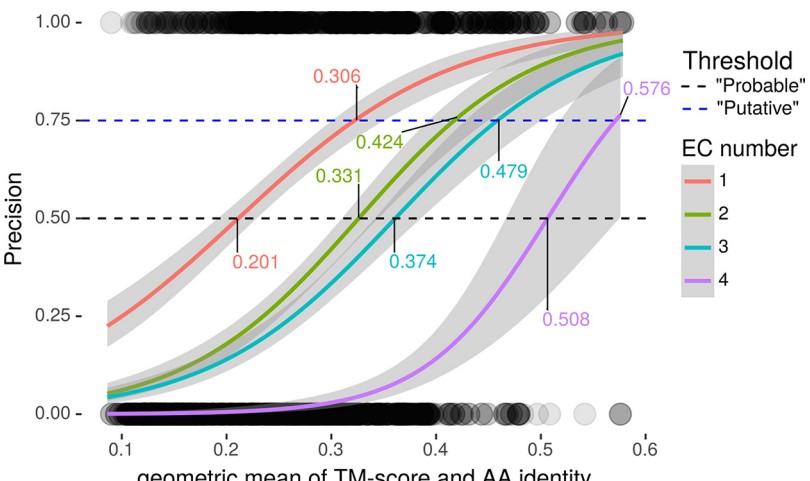

**FIG 1** Determining similarity thresholds for annotation inclusion criteria. Precision of EC number as regressed against the geometric mean of TM-score and AA% ($\mu_{geom}$) for each specificity tier. Horizontal lines define (50% and 75%) thresholds, the points where precision intersects with regression lines for each EC specificity curve (labeled). Circles at the bottom and top are individual data points (incorrect = 0 and correct = 1; *y* axis, precision; *x* axis, $\mu_{geom}$). Circles are rendered at 10% opacity to visualize observation density. Only templates with AA% of <40% were included.

"probable" as qualifying adjectives (Fig. 1). Through this procedure we defined distinct thresholds for ascribing "putative" or "probable" status to enzymatic function at each of the 4 tiers of EC specificity. We incorporated EC numbers and GO terms from similar structures deposited in Protein Data Bank (PDB) hierarchically, according to evidence reliability (Text S1 and Fig. S3). After the quality control pipeline described below, we recorded annotations in NCBI Table File Format and according to GenBank Prokaryotic Annotation Guide (www.ncbi.nlm.nih.gov/genbank/genomesubmit_annotation/) syntax and guidelines (integrated with manual curations from literature) and collated them into a unified functional annotation in GFF3 format (Fig. 2).

Although using $\mu_{geom}$ to determine inclusion criteria is useful for proteins with PDB entries of somewhat homologous sequence, it would not capture relationships by structural analogy or remote homology (because their low AA% would lower their score). To identify potential analogs and remote homologs, we used "$TM_{ADJ}$," an adjusted TM-score that accounts for model quality to conservatively estimate the TM-score between the true structure of a modeled protein and its putative homolog/analog of solved structure (Materials and Methods). We reexamined hits with $TM_{ADJ}$ values that indicated matching topology according to previous benchmarks (21) (Text S1) and annotated function with EC numbers, GO terms, and product names (Text S1 and Materials and Methods).

**Validating structure-based annotations.** To validate our structure-based functional inference approach, we ran proteins with annotations derived only from structural similarity (*n* = 366) through HHpred (43) (Fig. 3), a server that detects remote homology between proteins by comparing hidden Markov model profiles (43). We compared enzymatic structure-derived annotations (those with EC numbers, *n* = 335 distinct EC number annotations from 271 proteins) programmatically and nonenzymatic annotations manually (*n* = 95, Data Set S3 and Materials and Methods). Evaluating only the annotations to at least the second EC number level (*n* = 325), most structure-inferred predictions were partially (288/335, 86.0%) or wholly (266/335, 79.4%) corroborated by HHpred (Fig. 3C), substantiating the validity of our structure-based approach to functional inference. Partially corroborated annotations (e.g., 3.1.2.4 to the level of 3.1.2.- but not the fourth level of EC specificity) were revised to reflect the less specific, HHpred-supported level of functional detail and manually reconciled in cases where multiple EC numbers were corroborated (Materials and Methods).

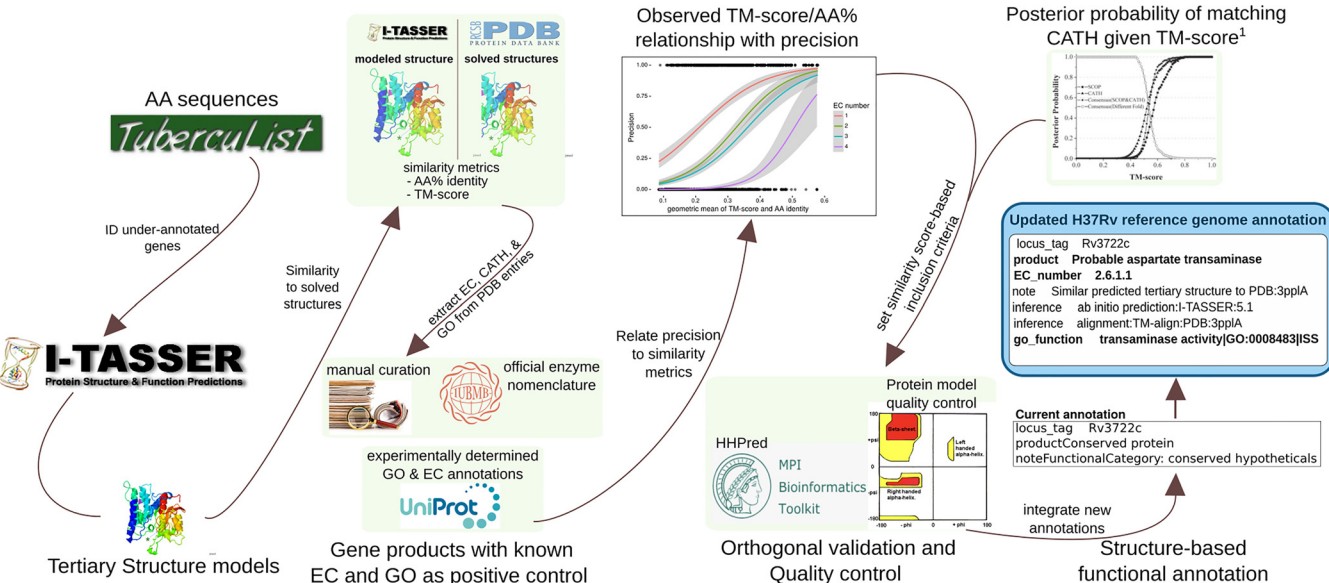

**FIG 2** Information flow for producing annotations from structural similarity. The flow of information and procedures for acquiring, processing, filtering, and representing information, running from retrieval of amino acid sequences to the final updated H37Rv annotation. Some details are omitted for clarity. The 1,725 amino acid sequences were retrieved from TubercuList and run through a local installation of I-TASSER v5.1. Of 1,725 amino acid sequences, 1,711 had models generated successfully. Comparison metrics for sequence (amino acid identity) and structure (TM-score) were extracted from I-TASSER output. To set criteria for annotation transfer, precision (equation 1) of GO Term and EC number concordance between similar matches on PDB and true function of 363 positive controls with GO terms and EC numbers of known function were regressed against extracted similarity metrics to generate a curve relating the geometric mean of TM-score and amino acid similarity to precision. These informed inclusion thresholds for transferring GO and EC annotations from structures on PDB similar to the 1,711 modeled structures. CATH topology folds were transferred according to a previous precision curve based on TM-score. This threshold was also used for inclusion of protein classes that vary in sequence more than structure (e.g., transporters) and as criteria for transferring annotations from structures that were not annotated with EC numbers or GO terms. Annotations derived only from structure were passed through orthogonal validation and manual structure analysis for verification that transferred annotations were reasonable. All annotations were programmatically collated into an updated H37Rv reference genome annotation.

Next, we assessed protein structure model quality using the fraction of residues in "most favored" regions of Ramachandran plots (Materials and Methods). Screening for abnormally low fractions can identify models with sterically untenable residue configurations, signaling low model quality (44). A threshold of 90% is often used for solved proteins (45), but we expected deviation from 90% even in quality models (since they are models rather than solved structures). To determine an acceptable threshold, we compared the distribution of residue fractions in "most favorable" regions among models with functions fully corroborated by HHpred with that of 29 models wholly uncorroborated by HHpred. Fractions for HHpred-corroborated proteins distributed unimodally and peaked around 90% of residues falling in the "most favorable" region (median = 89.15%). This observation is consistent with HHpred-corroborated proteins having high-quality structures and informs us of the range of fractions characteristic of high-quality structural models. Models with functions wholly uncorroborated by HHpred, meanwhile, distributed bimodally, with one mode resembling the fully corroborated distribution and the second mode peaking at a lower fraction (Fig. 3D). This bimodal distribution is consistent with a mixture of quality models and truly poor models. To distinguish between poor- and high-quality models in the wholly uncorroborated set, we implemented a heuristic threshold at the intersection of the two distributions (75%, Fig. 3D). After removing models below the threshold, the remaining uncorroborated structures formed a single peak that resembled the HHpred-corroborated proteins (Fig. 3E). We used this threshold (75%) as the minimum acceptable fraction for HHpred-uncorroborated proteins to be considered for structure-based functional annotation.

Seven of the protein models with exclusively wholly unsupported structure-based annotations (*n* = 29) were PE_PGRS protein models that resembled fatty acid synthase (FAS) subunit protein structures (particularly *Saccharomyces cerevisiae* PDB template 2pff). All seven failed Ramachandran filtering. This underscores the importance of these

mSystems®

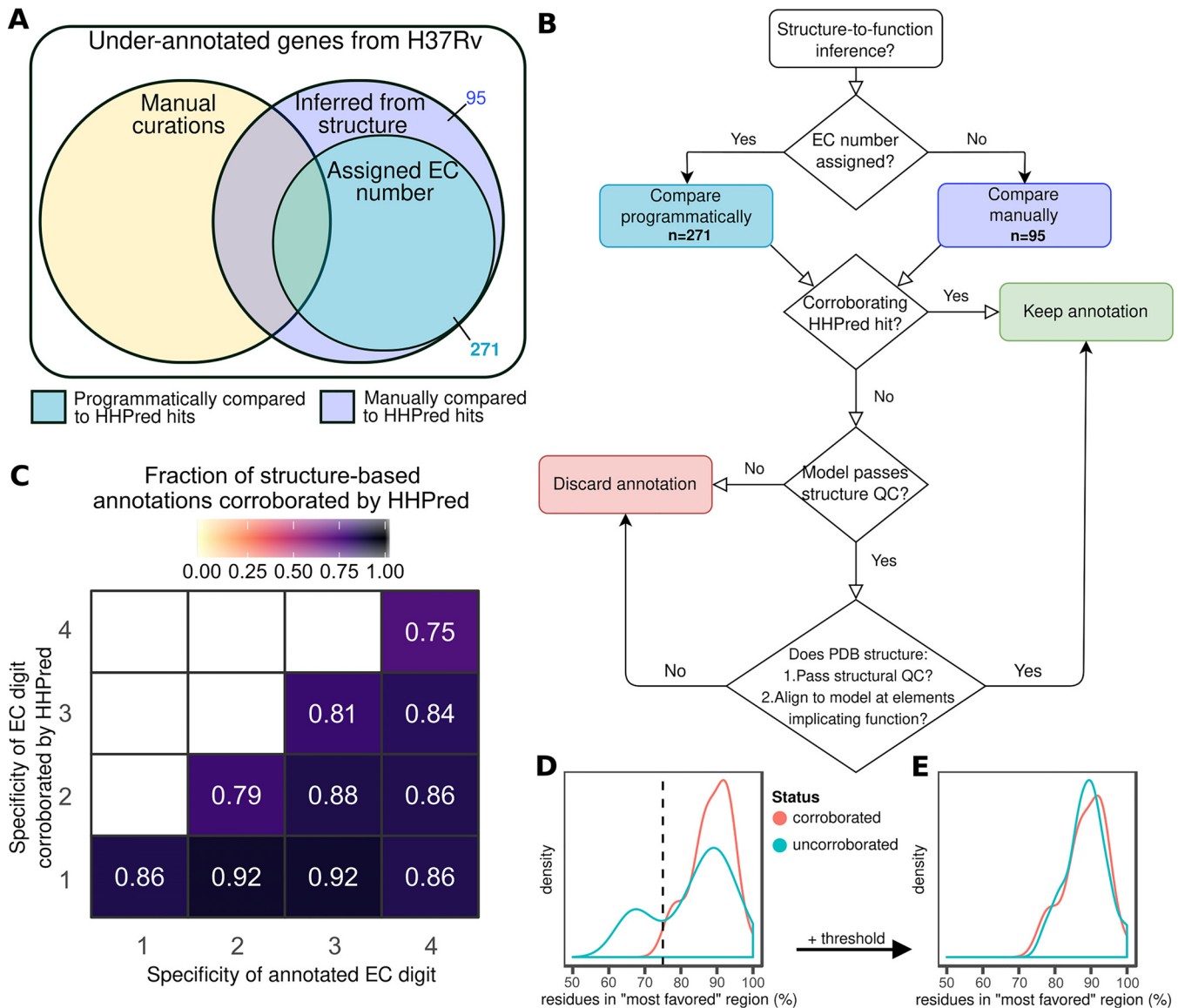

**FIG 3** Orthogonal validation and quality assurance for structure-to-function inference. (A and B) Quality assurance and validation protocol assignment (A) and decision workflow for retaining functional annotations inferred from structural similarity to proteins of solved structure and known function (B). (C) Heat map depicting fraction of EC number inferences corroborated by HHpred at each level of specificity. Fraction denominator is binned according to the number of EC digits annotated (*x* axis). (D and E) Structure quality assurance. Distribution of fraction of residues in "most favorable" region of Ramachandran plot prior to (D) and following (E) application of a heuristic threshold to discard biophysically improbable structural models.

quality control (QC) steps and suggests they excluded models implicating false functional analogies as intended. These annotations were likely artifactual, owing to glycine-abundant, low-complexity regions of PE_PGRS proteins aligning to the hydrophobic regions of large eukaryotic synthases, inflating their similarity score and spuriously implying structural similarity.

Since HHpred is designed to detect homology between proteins (43) (but not necessarily analogy—though analogous hits can arise), there may be genuine functions inferred by our structural similarity pipeline that HHpred did not corroborate. To preserve such annotations while ensuring annotation quality, we manually inspected HHpred-uncorroborated annotations (Fig. 3B) for protein models that passed Ramachandran filtering (*n* = 22). To accept annotations, we verified template protein quality, structural alignment of regions underlying function, and conservation of structural features and key functional residues. This step salvaged structure-derived functional annotations for nine proteins (Table 3 and Data Set S3), two of which we highlight in detail in Fig. 4.

**TABLE 3** Protein functions inferred by structural similarity and confirmed through manual structural analysis[a]

| Rv no. | PDB ID | AA % | Inferred EC | Updated EC | Change | Recommended product name | Product name (Mycobrowser) |
|---|---|---|---|---|---|---|---|
| Rv0036c | 2nsg | 14% | 5.2.1.- | None | Reduced specificity | Putative thiol-dependent DinB-like metalloenzyme | Conserved protein |
| Rv0738 | 2nsg | 17% | 5.2.1.- | None | Reduced specificity | Putative thiol-dependent DinB-like metalloenzyme | Conserved protein |
| Rv1632c | 2p12 | 59% | 3.6.1.- | 3.6.1.- | Verified | Putative cytidylyl-2-hydroxypropylphosphonate hydrolase | Hypothetical protein |
| Rv1727 | 2nsg | 12% | 5.2.1.- | None | Reduced specificity | Putative thiol-dependent DinB-like metalloenzyme | Conserved hypothetical protein |
| Rv1734c | 3l60 | 26% | 1.2.4.- | 2.3.-.- | Changed function | Putative acyltransferase | Conserved hypothetical protein |
| Rv1775 | 3hwp | 29% | 3.-.-.- | 3.7.1.- | Increased specificity | Putative 3-oxo-carboxylic acid hydrolase | Conserved hypothetical protein |
| Rv2036 | 2nsg | 15% | 5.2.1.- | None | Reduced specificity | Putative thiol-dependent DinB-like metalloenzyme | Conserved hypothetical protein |
| Rv2968c | 3kp9 | 21% | 1.1.4.- | 1.17.4.- | Nomenclature change | Putative vitamin K epoxide reductase | Probable conserved integral membrane protein |
| Rv3224B | 1dbx | 38% | 3.1.1.- | None | Reduced specificity | Putative *trans*-editing enzyme | Conserved hypothetical protein |

[a]Proteins with functions transferred from top PDB matches to their tertiary structural models subsequently verified through manually inspecting functionally essential protein features. Proteins shown lacked functional annotation output by HHpred corroborating the function from the structure-to-function annotation pipeline. Most had the same PDB template most closely matching the I-TASSER modeled structure among the top hits in HHpred but without functional information output and typically among dozens of other hits in PDB with similarly high match confidence uncorroborated by product function from HHpred.

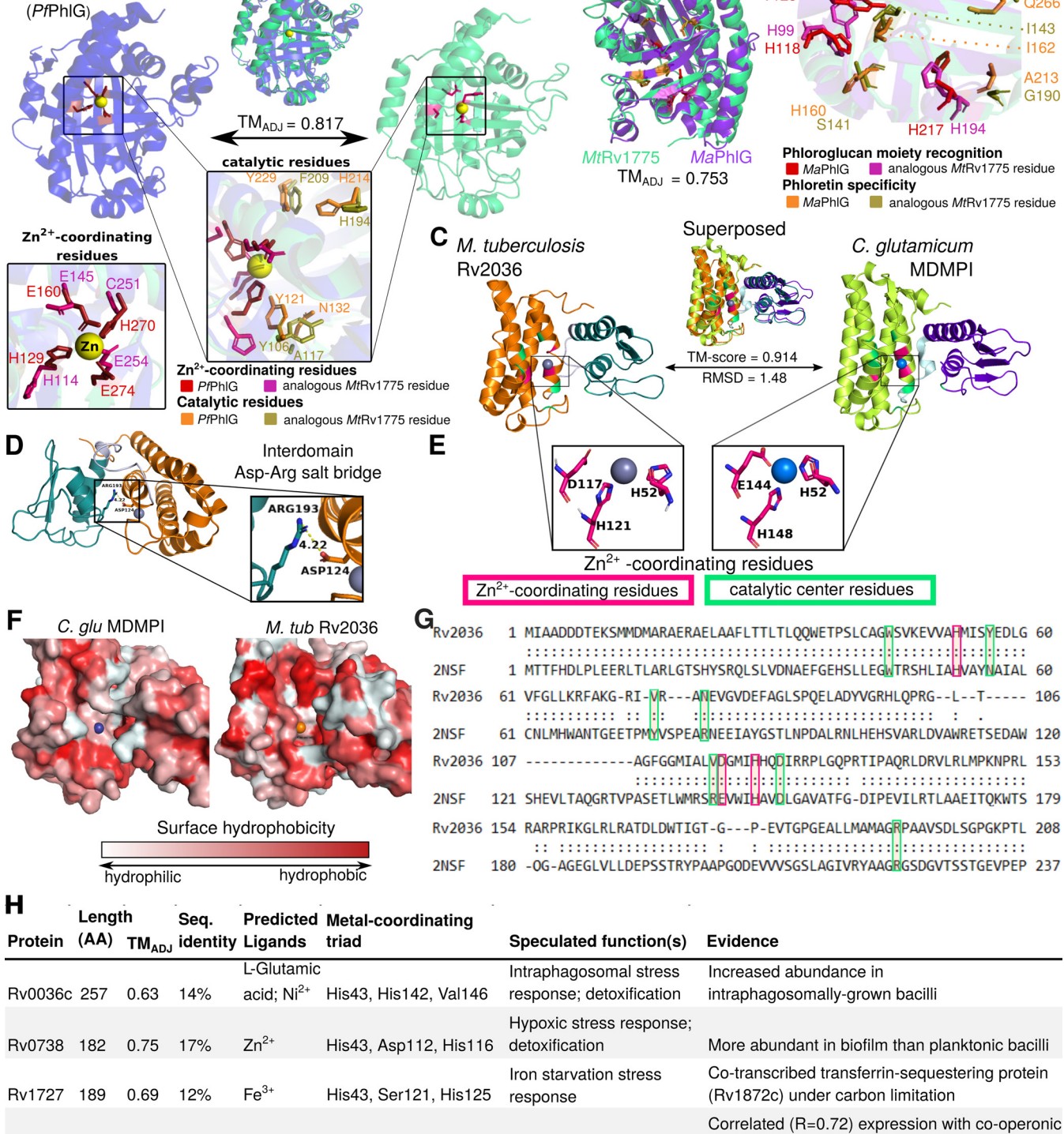

**FIG 4** Manual structural analysis refines functional annotations uncorroborated by HHpred. (A and B) Conservation of structure and sequence features essential for C—C bond hydrolysis supports the inferred hydrolase function of Rv1775. (A) Structural alignment of modeled Rv1775 and its closest structural match (PDB ID 3hwp), a 2,4-diacetyl-phloroglucinolhydrolase of *Pseudomonas fluorescens* (*Pf*PhlG), The structures are superposed (top). Zoomed and reoriented images of *Pf*PhlG zinc-coordinating (box on left) and catalytic (popout) residues superposed with analogous *Mt*Rv1775 residues. (B) Comparison of functional and structural features between *Mt*Rv1775 and a putative PhlG homolog of *M. abscessus* (*Ma*PhlG), phloretin hydrolase, which catalyzes C—C bond hydrolysis of a different substrate. Comparison carried out in a similar scheme as in panel A. Superposition of the putative homologs, color annotated with conserved residues essential for phloroglucol moiety recognition and for phloretin substrate specificity in *Ma*PhlG (47). The structural

In the first example (Fig. 4A and B), manual structural analysis fully corroborates the HHpred-uncorroborated function inferred from structure and extends annotation specificity. Originally, Rv1775 was ascribed putative hydrolase function by our structure-function inference pipeline. Its structural model is globally similar (TM$_{ADJ}$ = 0.817) to 2,4-diacetylphloroglucinol hydrolase PhlG (EC 3.7.1.24) from *Pseudomonas fluorescens* (46) (PfPhlG) despite only modest sequence similarity (27.6%). Comparison of Rv1775 to PfPhlG (Fig. 4A) and potential mycobacterial homolog (TM$_{ADJ}$ = 0.753, AA% = 28.4) phloretin hydrolase (EC 3.7.1.4) of *Mycobacterium abscessus* (47) (MaPhlG) showed conserved Zn$^{2+}$-coordinating and catalytic residues in the Rv1775 protein model (Fig. 4A and B). These conserved features suggest Rv1775 encodes a hydrolase acting on C—C bonds (EC 3.7.-.-), an uncommon class of catalytic activity (46). The only subsubclass within EC 3.7.-.- is 3.7.1.-, suggesting Rv1775 is a 3-oxoacid carboxylase.

Although the precise substrate(s) of Rv1775 is indiscernible from structural comparison alone, examining its structure suggests a potential role in lipid metabolism. It shares the phloroglucinol moiety recognition residues conserved across R-phloroglucinol hydrolases but lacks conserved residues required for phloretin hydrolysis (Fig. 4B). This suggests Rv1775 is not a phloretin hydrolase but may act on substrate(s) containing a phloroglucinol moiety or similar aromatic chemical species. Considering reports of *M. tuberculosis* utilizing cholesterol as a carbon source (48), known C—C hydrolytic enzymes in *M. tuberculosis* cholesterol catabolism (49), and gaps in the current understanding of cholesterol catabolism (50), cholesterol ring species are plausible C—C hydrolysis substrates.

In the second example (Fig. 4C to H), we examine one of four HHpred-uncorroborated proteins structurally resembling mycothiol-dependent maleylpyruvate isomerase (MDMPI, a DinB superfamily protein; PDB accession no. 2nsf) of *Corynebacterium glutamicum* (*C. glu* MDMPI). This example illustrates the case when manual inspection corroborates conserved structural features yet precise molecular function remains indiscernible. Manual structural analysis of the putative MDMPI homologs validated that—despite low sequence homology (12 to 17% similarity)—structural features characteristic of DinB-like enzymes are conserved (shown for Rv2036, Fig. 4). All four putative DinB-like enzymes were highly structurally similar to *C. glu* MDMPI (TM$_{ADJ}$ = 0.63 to 0.75, Fig. 4C) with a conserved hydrophilic core (Fig. 4F), predicted metal-binding sites (Fig. 4E), retained catalytic triad residues (51) (Fig. 4G), and conserved residues that form a salt bridge between the C- and N-domains (51) of MDMPI (Fig. 4D). However, DinB superfamily proteins comprise several functions (52), making even putative inference of a precise molecular function challenging. Most functionally characterized bacterial DinB-like enzymes are thiol dependent (52), and the putative MDMPI homologs' closest structural match was a mycothiol-dependent DinB-like enzyme, suggesting thiol dependence of these four proteins is probable, likely with mycothiol as the thiol cofactor (the predominant mycobacterial low-molecular-weight thiol). We annotated these genes as "putative thiol-dependent DinB-like metalloenzymes" and note as "potential (myco)thiol-dependent S-transferase (EC 2.-.-.-)" (53). For such cases, where structural modeling confidently ascribes protein family and features of structure but not function, integrating knowledge of the function of structural orthologs,

**FIG 4** Legend (Continued)

similarity and conserved zinc-coordinating and catalytic residues affirm Rv1775 as a bona fide C—C hydrolase, potentially with a substrate that includes a phloroglucol moiety but likely not phloretin. Conservation of structure and sequence features characteristic of DinB-like metalloenzymes exemplified by structural homology of Rv2036 and a mycothiol-dependent maleylpyruvate isomerase from *Corynebacterium glutamicum* (*C. glu* MDMPI) (C to G). (C) Superposition of Rv2036 structure model and *C. glu* MDMPI (PDB ID 2nsf). Conserved Zn$^{2+}$-coordinating (pink) and catalytic (green) residues are highlighted. (D) Highly conserved residues Arg$^{222}$ (C-terminal domain, Arg$^{193}$ in Rv2036) and Asp$^{151}$ (N-terminal domain, Asp$^{124}$ in Rv2036) are in close proximity (4.22 Å), suggesting conservation of their proposed role as interdomain protein stabilizers (51). (E) Spatial conservation of Zn$^{2+}$-coordinating residues of the catalytic triad (Asp and Glu are observed interchangeably) is consistent with conserved catalytic function. (F) Surface hydrophobicity of Rv2036 model and 2nsf shows that the hydrophilic core proposed to underlie MDMPI catalysis (51) is relatively conserved. (G) Structure-based sequence alignment of Rv2036 and *C. glu* MDMPI with conserved residues was manually annotated according to prior work (51). (H) Summary of relevant genomic context potentially informative of function, protein similarity metrics between putative *M. tuberculosis* MDMPI homologs and *C. glu* MDPMI, and predicted protein features. All structural images were rendered in PyMOL. Structurally homologous sequence alignments are based on TM-align (22) (**, <5 Å between residues; *, <10 Å between residues).

expression data, and genomic context can inform rational speculation about their function (Fig. 4H and Text S1).

**Hundreds of annotations inferred by structural similarity.** Our structural annotation pipeline inferred function from structure for 400/1,725 underannotated genes (23.2%, Data Set S1). Structure-derived annotations (mean C-score = 0.39) came from higher-quality models ($P = 1.83 \times 10^{-163}$, Student's *t* test) than proteins without passing matches (mean C-score = $-1.91$), and more specific annotations tended to come from higher-quality models (Text S1 and Fig. S5). Structure-based annotation captured putatively shared function for numerous previously unannotated proteins lacking appreciable sequence similarity (Table 4 and Data Set S3).

These remote homologs and structural analogs include an integral membrane methyltransferase, which can modify mycolic acids embedded in the *M. tuberculosis* cell wall essential for virulence (54) and redox response-related functions (Rv0052 and Rv3192) critical for enduring host immune defenses in macrophages (55).

**Putative efflux and transport proteins uncovered through structural similarity.** Membrane-spanning regions of transport proteins vary in sequence relative to structure far more than globular proteins (12, 56), making them good subjects for structure-based functional inference. Twenty-four proteins were identified as transport proteins and corroborated by HHpred (Data Set S3), including several matches with drug transport proteins (*n* = 8). Eight HHpred-corroborated proteins were not annotated with any transport function in UniProt (Table 5). Rv1510 and Rv3630 exclusively match drug transporters and are uncharacterized across functional databases. Rv3630 mutations have been reported in pyrazinoic acid (POA)-resistant mutants, but no clear causal link was identified (57). Rv1510 is a *Mycobacterium tuberculosis* complex marker in diagnostic assays (58), and its loss of function induces autophagy (59), suggesting Rv1510 is an autophagy antagonist important for human-adapted tuberculosis. Verapamil, a potent efflux pump inhibitor, induces autophagy (60), consistent with the putative function of Rv1510 in drug efflux, which could contribute to drug tolerance (58). These putative transporters might contribute to intrinsic efflux-mediated drug resistance and tolerance in *M. tuberculosis* (11). Other putative novel transport proteins may serve important homeostatic roles in the dynamic host microenvironment (61, 62) and could make attractive drug (63) and vaccine (64) targets.

**An updated *M. tuberculosis* reference genome functional annotation.** Through manual curation (*n* = 282) and structural inference (*n* = 400), we annotated 623 gene products, reducing underannotated genes by 36.1%. Including annotated CATH (Class, Architecture, Topology, and Homologous superfamily) topologies, functional notes, and ligand-binding sites (LBS) results in a total of 940 (54.5%) with original annotation (Fig. 5B). For genes lacking specific product annotations, CATH (Data Set S3L) and LBS assignments (Data Set S3D) can refine functional hypotheses and, in some cases, imply function directly (65). Tetracycline repressor folds (*n* = 17, Data Set S3M), for instance, function nearly exclusively as concentration-dependent transcriptional activators and vary in sequence yet are structurally homogeneous (66). CATH annotations were not used to inform product annotations nor to assign EC numbers in this annotation, however.

Our updated annotation provides function for 34.4% (45/131) of genes with hypothetical function identified in a recent systems resource as broadly conserved across mycobacteria (67) (Data Set S3 contains the full set). Mycobacterial core genes annotated include functions well established experimentally, such as essential component of the mycobacterial transcription initiation complex RbpA (https://gitlab.com/LPCDRP/Mtb-H37Rv-annotation/-/blob/master/features/Rv2050.tbl) and others not evident from extant literature but of potential clinical relevance, like the host-directed effector function inferred for Rv3909 (https://gitlab.com/LPCDRP/Mtb-H37Rv-annotation/-/blob/master/features/Rv3909.tbl). These annotations came in similar numbers from published experimental evidence (*n* = 21) and structural inferences (*n* = 24).

Updated annotations distribute across all segments of the chromosome (Fig. 5A) and implicate efflux proteins (Table 5), metabolic functions (Fig. 6), virulence factors, and functions key to survival during infection (Table 6) and under drug pressure (Table 7). Yet,

**TABLE 4** Novel annotations transferred through structural similarity despite low sequence similarity[a]

| Rv no. | Top I-TASSER hit | AA% | TM$_{ADJ}$ | PDB ID | Final annotation | Mycobrowser | UniProt | Mtb Network Portal | Type |
|---|---|---|---|---|---|---|---|---|---|
| Rv1139c | Integral membrane methyltransferase | 18 | 0.86 | 4a2n | Putative integral membrane methyltransferase | Conserved hypothetical membrane protein | Conserved hypothetical membrane protein (membrane protein) | None | Novel |
| Rv1766c | Copper-sensing transcriptional repressor CsoR | 29 | 0.84 | 4m1p | Putative transcription factor | Conserved protein | Conserved protein | None | Novel |
| Rv3192c | 5,10-Methylenetetrahydromethanopterin reductase | 16 | 0.83 | 1z69 | Putative monooxygenase | Conserved hypothetical alanine- and proline-rich protein | Conserved hypothetical alanine- and proline-rich protein | Oxidoreductase | More specific |
| Rv2141c | M20 family metallopeptidase | 20 | 0.82 | 2pok | Putative linear amide hydrolase | Conserved protein | Conserved protein | FIG016551: putative peptidase | Affirmatory |
| Rv1775 | 2,4-Diacetylphloroglucinol hydrolase | 29 | 0.82 | 3hwp | Putative 3-oxo-carboxylic acid hydrolase | Conserved hypothetical protein | Uncharacterized protein | None | Novel |
| Rv0052c | Isonitrile hydratase | 33 | 0.81 | 3noo | Putative hydrolyase/putative deglycase | Conserved protein | Conserved protein | ThiJ/PfpI family protein | Novel |
| Rv2036 | Mycothiol-dependent maleylpyruvate isomerase | 15 | 0.73 | 2nsg | Putative thiol-dependent DinB-like metalloenzyme | Conserved hypothetical protein | DinB family protein | None | More specific |

[a]Selected proteins with modeled structures highly similar to solved PDB structures of known function. Sequence similarities range in the "twilight zone" of sequence similarity, below which remote homology is undetectable by sequence similarity (132). A TM$_{ADJ}$ above 0.52 indicates that the template and the underannotated gene share structural folds. Annotations from UniProt, Mtb Network Portal, and TubercuList are shown, along with the highest error-adjusted structural similarity match, its identifier ("PDB"), and final product annotation. "Affirmatory" indicates corroboration of the annotations in UniProt or Mtb Network Portal. "Novel" annotations are annotations entirely novel to those in UniProt and Mtb Network Portal, while "More specific" annotations are in accord with annotations in other databases but describe product function in greater detail.

**TABLE 5** Putative transport proteins[a]

| Rv no. | Product annotation | AA (%) | PDB macromolecule name | PDB ID | TM-score | $TM_{ADJ}$ |
|---|---|---|---|---|---|---|
| Rv1085c | Putative membrane transporter receptor protein | 0.073 | Chloride-pumping rhodopsin | 5b2nA | 0.71 | 0.59 |
| | | 0.058 | Sodium-pumping rhodopsin | 4xtlA | 0.71 | 0.59 |
| Rv1462 | Putative transporter | 0.173 | ABC transporter, ATP-binding protein | 4dn7A | 0.79 | 0.67 |
| Rv1510 | Putative Na⁺/H⁺ antiporter drug efflux protein | 0.104 | Putative drug/sodium antiporter | 4z3nA | 0.89 | 0.60 |
| | | 0.088 | Multiantimicrobial extrusion protein [Na(+)/drug antiporter] MATE-like MDR efflux pump | 3mktA | 0.84 | 0.54 |
| Rv1680 | Putative phosphonate transporter component | 0.151 | PhnD, subunit of alkylphosphonate ABC transporter | 3p7iA | 0.91 | 0.77 |
| | | 0.165 | Binding protein component of ABC phosphate transporter | 3n5lA | 0.89 | 0.76 |
| Rv2325c | Putative transport protein | 0.165 | Putative cobalt ABC transporter, permease protein | 5d3mD | 0.87 | 0.60 |
| | | 0.215 | Energy-coupling factor transporter transmembrane protein EcfT | 4huqT | 0.81 | 0.54 |
| Rv2508c | Putative MFS membrane transporter | 0.104 | Solute carrier family 2, facilitated glucose transporter member 3 | 5c65A | 0.79 | 0.63 |
| | | 0.119 | D-Xylose-proton symporter | 4gbyA | 0.78 | 0.63 |
| Rv3630 | Putative Na⁺/H⁺ antiporter drug efflux protein | 0.087 | Multiantimicrobial extrusion protein [Na(+)/drug antiporter] MATE-like MDR efflux pump | 3mktA | 0.84 | 0.63 |
| | | 0.108 | Putative drug/sodium antiporter | 4z3nA | 0.83 | 0.62 |

[a]Matches between proteins encoded by underannotated genes (Locus) and transport protein structure entries in Protein Data Bank (PDB). Only matches undescribed as transport proteins on UniProt are included (see Data Set S3E for all such matches). The top two matches are shown, if they exceed the adjusted TM-score ($TM_{ADJ}$) of >0.52 (the TM-score corresponding to matching topologies >50% of the time). AA% refers to the amino acid identity shared between the aligned region of the protein in *M. tuberculosis* and its match on PDB. MFS, major facilitator superfamily; MDR, multidrug resistance.

many underannotated genes remain without products or functional notes assigned ($n = 785$). Of these 785 remaining underannotated genes (Data Set S1), 190 have quality models (C-score $> -1.5$) but lack annotations meeting inclusion criteria. Meanwhile, 182 of those remaining have product annotations in UniProt or Mtb Network Portal. Remaining still, however, are 466 underannotated genes with neither quality structure nor functional annotation in these databases. These genes frequently cluster consecutively along the genome (99 genes across 23 clusters, Data Set S1G), forming syntenic blocks of unknown function. Genomic context suggests several of these clusters have roles in virulence and drug tolerance (Data Set S1G).

Genes remaining without any form of annotation (Fig. 5D) were overrepresented ($P = 0.0011$, odds ratio = 1.35, Fisher's exact) near the terminus ($\pm 250$ kb from half the genome length) of the chromosome (*ter*-proximal genes, Data Set S1H). An even stronger bias for uncharacterized genes can be seen for genes transcribed opposite the direction of replication ($P = 1.14 \times 10^{-7}$, odds ratio = 1.53; Fisher's exact). To ensure that circumstantial factors such as PE/PPE or insertion element density were not accounting for the apparent orientation and spatial trends across the chromosome, we removed all PE/PPE and insertion sequence and phage genes and repeated the analysis. The trend strengthened for the *ter*-proximal gene bias ($P = 0.0034$, odds ratio = 1.44; Fisher's exact) and decreased only marginally ($P = 2.53 \times 10^{-6}$, odds ratio = 1.49; Fisher's exact) for the orientation bias.

These biases are consistent with three previously noted trends that could influence the likelihood of gene characterization. First is the general trend of decreased gene expression as a function of distance from the *oriC* in bacteria (68). On average, highly expressed genes are more amenable to functional characterization. Second is the strong bias for symmetric inversions around the terminus (69), particularly in *Actinobacteria* (70). Hypotheses leading to experimentally determined functions are often informed by orthology, which can be inferred by conserved synteny between species (71). Common inversions around the terminus can disrupt this synteny with increased frequency. Disruption

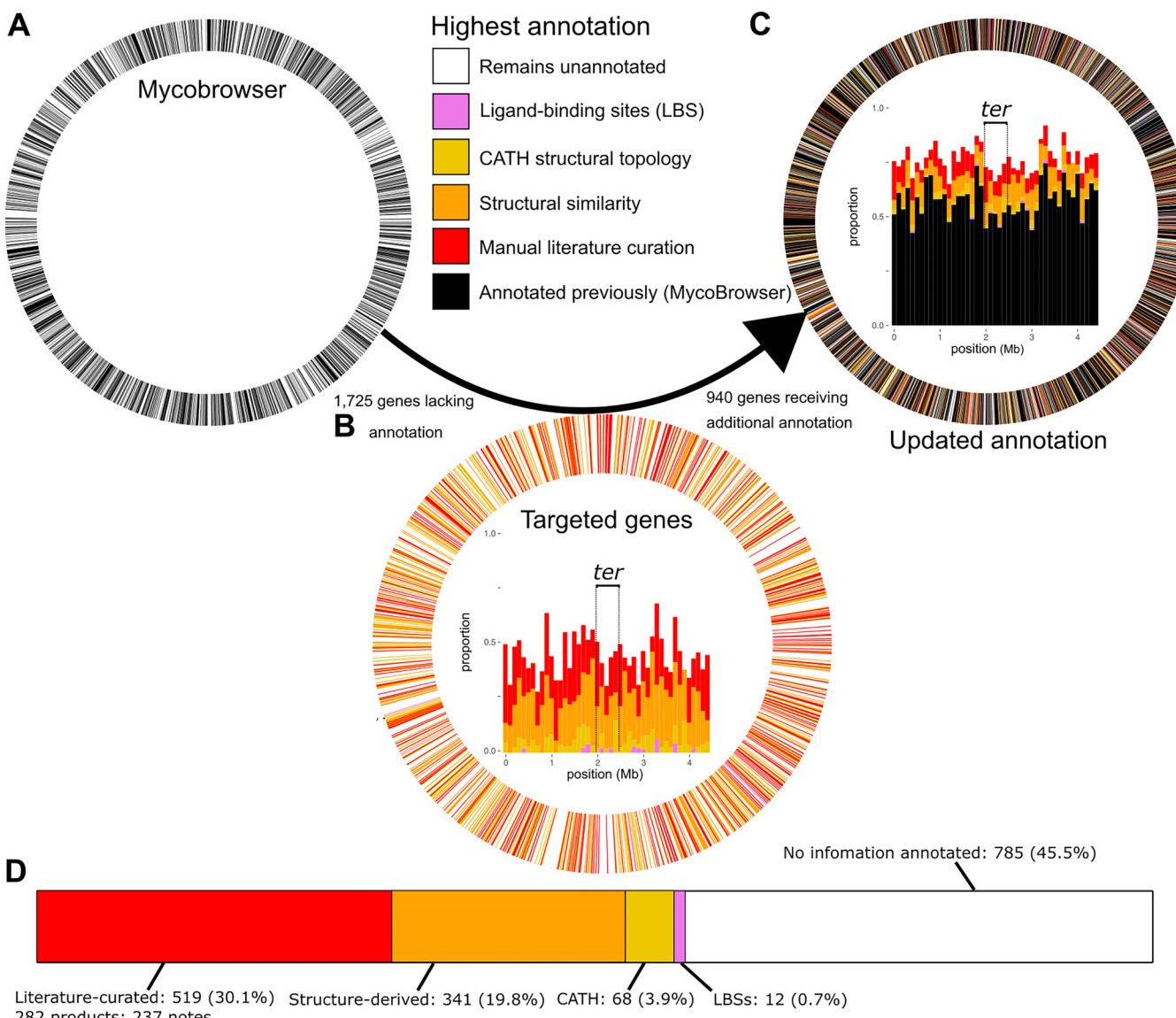

**FIG 5** An updated H37Rv functional annotation. (A to C) Circos plots illustrating annotation coverage prior to the annotation effort (left) and following it (right), colored according to annotation status. In plots A and C, all 4,031 CDS are represented as segments of equal width whereas plot B segments the ring into only the 1,725 underannotated genes. Black genes reflect what was on TubercuList, are considered "annotated," and are mutually exclusive from the 1,725 underannotated genes (white). Panel B shows only the 1,725 underannotated genes, whereas panels A and C include all 4,031 original CDS. Inside the Circos rings are stacked bar charts with genes in 100-kb bins according to gene start position. The terminal-proximal (±250-kb) region is marked with dashed lines and labeled (*ter*). (D) Cumulative number of genes annotated, by annotation type. LBS, ligand binding site. Percentages refer to underannotated genes annotated/1,725 initial underannotated genes. Genes are binned into mutually exclusive categories hierarchically: manually curated product name > structure-derived > literature notes > CATH > LBS. Manually curated and literature note categories are combined as "Literature-curated" in the visualization. For the purposes of these counts, functional notes from publications implicating many proteins but not clearly establishing function were not counted (e.g., references 32, 103, and 125).

can occur globally—through moving across the chromosome by inversion—and locally, by inversion boundaries interrupting operons or other syntenic features. Third, genes transcribed opposite the direction of replication frequently collide with the replication machinery, making them more mutable than genes with transcription and replication cooriented (72). This increases the likelihood of weakened promoters or loss-of-function mutations evolving *in vitro* for genes nonessential in H37Rv. One potential confounder is that genes encoding virulence/toxin proteins are enriched on the lagging strand (72). As these genes operate in the context of infection, they are challenging to functionally characterize, which may contribute to the observed enrichment of uncharacterized genes on the lagging strand.

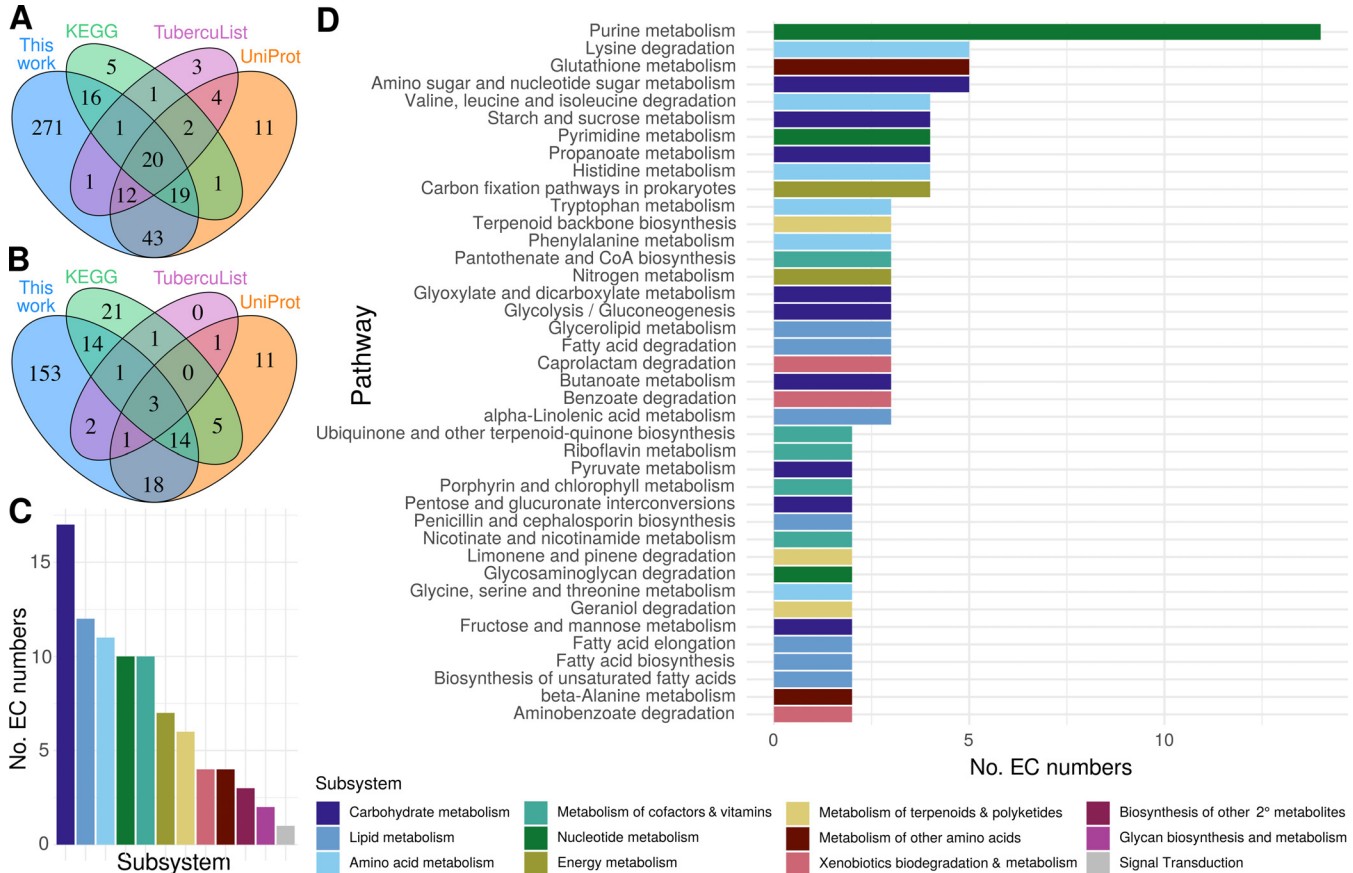

**FIG 6** Functional annotations across *M. tuberculosis* metabolism. Annotated EC numbers for manually curated and structure-inferred products were compared with annotations for each underannotated gene in popular databases. (A) Set analysis of underannotated genes (UAG) with an EC number assigned in this study compared to popular databases. (B) Novelty of EC numbers for UAG annotated in this study with respect to popular databases. (C and D) Distribution of EC numbers annotated across KEGG subsystems (C) and pathways (D). Generic KEGG subsystems are depicted. All pathways with at least three genes have the number of EC numbers displayed. For subsystems with no pathways with three or more genes, the highest total pathway is displayed.

Turning our attention to metabolism, 381 underannotated genes were annotated with EC numbers (Data Set S1I and Materials and Methods), over two-thirds of which were absent from other databases (Fig. 6A and B). Fully specific (fourth EC digit) EC numbers (*n* = 92) were ascribed to 85 genes. These newly annotated reactions span diverse metabolic pathways and subsystems (Fig. 6 and Data Set S1J), many implicated in mediators of *M. tuberculosis* virulence such as lipid and polyketide and terpene metabolism (73–75), which are integral to the unique composition of the mycomembrane. Proteins of these pathways have important immunity-subverting functions (76) at the host-pathogen interface (77). For instance, terpenes play an immunomodulatory role early in *M. tuberculosis* infection and phagosomal maturation (78–80), are potential agonists of antibiotics for TB treatment (81), and include cell membrane surface-expressed molecular species unique to *M. tuberculosis* (82). The numerous carbohydrate-metabolizing products (Fig. 6D) may identify alternative metabolic pathways in *M. tuberculosis* and aid in gap-filling efforts in *M. tuberculosis* metabolic reconstructions.

**Integration with recently published functional screens.** Next, we assessed how much novel functional information our annotation added to ambiguously or hypothetically annotated genes from a recent transposon mutagenesis study that sought to identify specific bacterial functions limiting drug efficacy during a mouse model of infection (83). We assessed two sets of genes identified in the study. In the first set of underannotated genes—those newly reported to as essential for optimal growth in mouse infection—one-third (23/69) could be updated by our annotations (Table 6). Fifteen were structural inferences, demonstrating the value of structure-based inference

**TABLE 6** Updated annotations add functional knowledge to genes required for optimal fitness during TB infection[a]

| Rv no. | Source annotation | New annotation | PubMed ID(s) | PDB ID(s) |
|---|---|---|---|---|
| Literature annotations | | | | |
| Rv1205 | Hypothetical protein | Riboside monophosphate phosphoribohydrolase | 25728768 | |
| Rv2018 | Hypothetical protein | Probable antitoxin VapB/antigen | 28066388, 23140854 | |
| Rv2272 | Transmembrane protein | Probable gamma delta T-cell activator | 23389928 | |
| Rv2525c | Tat pathway signal sequence | Probable peptidoglycan hydrolase | 16952959, 25869294, 25260828 | |
| Rv2923c | Hypothetical protein | Probable osmotically induced bacterial protein C (OsmC, a probable peroxide reductase) | 22088319 | |
| Rv3632 | Membrane protein | Putative flippase | 21030587 | |
| Rv3763 | Lipoprotein LpqH | Adhesin/antigen LpqH | 16098710, 12594264 | |
| Rv3788 | Hypothetical protein | Probable secondary channel binding factor of RNA polymerase | 22194445 | |
| Literature and structural inference | | | | |
| Rv0191 | Integral membrane protein | Putative efflux pump | 25690361, 12520088, 22132058 | 5c65, 4gby |
| Rv1433 | Exported protein | Probable L,D-transpeptidase LdtMt3 | 24041897 | 3tur, 3vae, 4jmn |
| Rv1769 | Hypothetical protein | T-cell antigen/putative aldehyde-lyase | 26853625, 15102765 | 4v15 |
| Rv3722c | Hypothetical protein | Probable serine hydrolase, probable aspartate transaminase | | 3ppl |
| Structural inference | | | | |
| Rv0047c | Hypothetical protein | Putative transcriptional regulator | | 1yg2a, 3l9f |
| Rv0259c | Hypothetical protein | Putative lyase | | 2jh3, 4ccs |
| Rv0323c | Hypothetical protein | Putative hydrolyase/putative linear amide hydrolase | | 1q7t, 5cgz |
| Rv0449c | Hypothetical protein | Putative oxidoreductase | | 2ive, 1sez, 3nks, 3i6d, 3lov |
| Rv0767c | Hypothetical protein | Putative transcription factor | | 3mnl |
| Rv1085c | Hemolysin-like protein | Putative membrane transporter receptor protein | | 5b2nA, 4xtlA |
| Rv2052c | Hypothetical protein | Putative endodeoxyribonuclease | | 3igh |
| Rv2160A | Hypothetical protein | Putative transcription factor | | 2hyj |
| Rv3226c | Hypothetical protein | Putative peptidase | | 2icu |
| Rv3433c | Hypothetical protein | Putative hydrolyase/putative isomerase | | 2ax3 |
| Rv3719 | Hypothetical protein | Putative amide-bond oxidoreductase | | 3dq0, 2exr, 4o95 |

[a]Source annotation is the annotation listed by Bellerose et al. (83) and new annotation derived from the current project. Protein Data Bank identifiers (PDB ID) of the protein structures matching H37Rv protein models are listed for structure-based annotations. PubMed IDs are listed for the papers from which functional annotations were manually curated.

of putative function where the difficulty of recapitulating complexities of the host environment challenges functional elucidation through experiment. Notably, following its inference based on structure, Rv3722c has since been confirmed to indeed encode an aspartate transaminase (84) and Rv1085c has been found likely not to encode hemolysin (85), substantiating the structure-derived functional annotations in Table 6.

Our annotation functionally described 13/27 underannotated genes affecting drug sensitivity (Table 7). Notably, some genes affecting drug sensitivity have published functions consistent with the mechanism of action of the drug of interest but listed without annotation. For instance, the authors noted cell wall permeability as a central theme among genes dictating sensitivity to rifampin (RIF), and disruption of Rv2190c—a peptidoglycan hydrolase—rendered mutants hypersusceptible to RIF, consistent with an effect on cell wall permeability. Others (e.g., Rv1184c) were unannotated in their primary data, but their functional ties were discussed in the text, suggesting the function was curated from literature. Our updated annotation centralizes such functional knowledge.

**Structural models enable functional interpretation of novel PZA-resistant mutants.** Next, we applied our annotations prospectively to a new resistance screen, querying the molecular basis of pyrazinamide (PZA) resistance in *M. tuberculosis*. PZA is

**TABLE 7** Updated annotation enriches functional interpretation of underannotated genes affecting drug sensitivity[a]

| Rv no. | Source annotation | New annotation | Enriched condition(s) | Evidence (PubMed ID[s]) |
|---|---|---|---|---|
| Rv0998 | Hypothetical protein | cAMP-dependent lysine acetyltransferase | EMB, RIF, HRZE | 23553634 |
| Rv1205 | Hypothetical protein | Riboside monophosphate phosphoribohydrolase | EMB (+) | 25728768 |
| Rv0767c | Hypothetical protein | Putative transcription factor | INH | 3mnl, 3bjb (PDB IDs) |
| Rv3131 | Hypothetical protein | Putative nitroreductase | INH | 27094446, 28261197 |
| Rv2140c | Hypothetical protein | Phosphatidylethanolamine-binding protein TB18.6 | INH | 23907008, 27895634, 26238929 |
| Rv2061c | Hypothetical protein | Probable serine hydrolase | INH | 26853625, 26536359 |
| Rv3267 | Hypothetical protein | Probable peptidoglycan-arabinogalactan ligase | INH | 27486192 |
| Rv1770 | Hypothetical protein | Probable serine hydrolase | PZA | 26853625 |
| Rv3005c | Hypothetical protein | Probable membrane oxidoreductase component (MRC) DoxX | RIF | 26067605 |
| Rv1184c | Exported protein | Mycoacyltransferase | RIF | 25331437, 25124040 |
| Rv3036c | Secreted protein | Secreted esterase | RIF, INH | 25224799 |
| Rv2190c | Hypothetical protein | Peptidoglycan peptidase RipC/antigen | RIF | 24843173, 22952680, 28241799 |
| Rv0079 | Hypothetical protein | Putative dormancy-associated translation inhibitor (DATIN) | RIF (+) | 22719925, 23819907, 28261197 |

[a]Source annotation is the annotation listed by Bellerose et al. (83). Enriched conditions are the drugs' exposure under which differential mutant abundance was observed. Sources of updated annotation are listed in the evidence column. INH, isoniazid; RIF, rifampin; EMB, ethambutol; PZA, pyrazinamide; HRZE, combination regimen of INH, RIF, PZA, and EMB. "(+)" indicates enrichment observed at multiple time points.

a cornerstone of modern tuberculosis therapy, yet the mechanism by which it exerts its antitubercular activity remains elusive. PZA is a prodrug that must be converted to its active form pyrazinoic acid (POA) by a mycobacterial amidase (86). While multiple explanations for POA action have been proposed (87–89), many of these models have not held up to scrutiny (90–92). Recently, several groups have shown that POA either directly or indirectly disrupts mycobacterial coenzyme A (CoA) biosynthesis (93–95). Identification of novel resistance mechanisms could shed additional light on the elusive action of this drug. Thus, a library of $10^5$ transposon-mutagenized *M. tuberculosis* H37Rv mutants was used to select for POA-resistant isolates. While the frequency of spontaneous resistance to POA is approximately $10^{-6}$, the frequency of resistance from our transposon-mutagenized library was $10^{-3}$. Four mutant strains chosen for further characterization of drug resistance profile and transposon insertion site (Fig. 7) showed insertions in genes of unknown function. Each of these strains showed ≥2-fold resistance to PZA and POA (Fig. 7A to D) and no change in INH susceptibility (Fig. 7A to D) compared to wild-type H37Rv.

To interpret how the interrupted genes might contribute to PZA resistance, we inspected the structural and functional data available from our I-TASSER results (Fig. 8 and 9). PZA is a structural analog of nicotinamide (96), suggesting the putative nicotinamide binding domain of Rv2705c (Fig. 7A) may interact directly with PZA or POA. While it remains difficult to confidently annotate Rv2706c (Fig. 7B), considering its position immediately upstream of Rv2705c, it may alter PZA sensitivity by influencing expression of Rv2705c. An alternative explanation that cannot be ruled out is that Rv2705c::Tn confers PZA resistance through its interruption of the N terminus of Rv2704 (Fig. 7A), a structurally solved YjgF superfamily protein (PDB ID 3I7T) with probable ester hydrolase function (Data Set S1).

Rv3256c structurally resembles multiple phosphosugar isomerases—particularly phosphoglucose (PGI) and phosphomannose (PMI)—and glutamine-fructose-6-phosphate transaminases (GlmS). Rv3256c has neither the conserved residues essential for PMI/PGI catalysis (97) (Fig. 8A and C) nor the glutamine amidotransferase domain required for GlmS activity (98) (Fig. 8B and D), effectively ruling out these functions. The common structural feature among these functionally disparate matches is a sugar isomerase (SIS) domain (Fig. 8). The SIS domain is a phosphosugar-binding module (99), implicating Rv3256c in phosphosugar metabolism or its regulation. Rv3256c lacks the helix-turn-helix (HTH) domain common to RpiR-like SIS domain proteins (99) that regulate phosphosugar metabolism genes, refuting the possibility of an RpiR-like transcriptional regulatory function. Flanking Rv3256c (Fig. 7C), however, are mannose donor biosynthesis genes—Rv3255c (a PMI) and Rv3257c (a phosphomannomutase). In *Mycobacterium smegmatis*, Rv3256c overexpression decreased cell surface mannosylation (100), consistent with a role in regulating

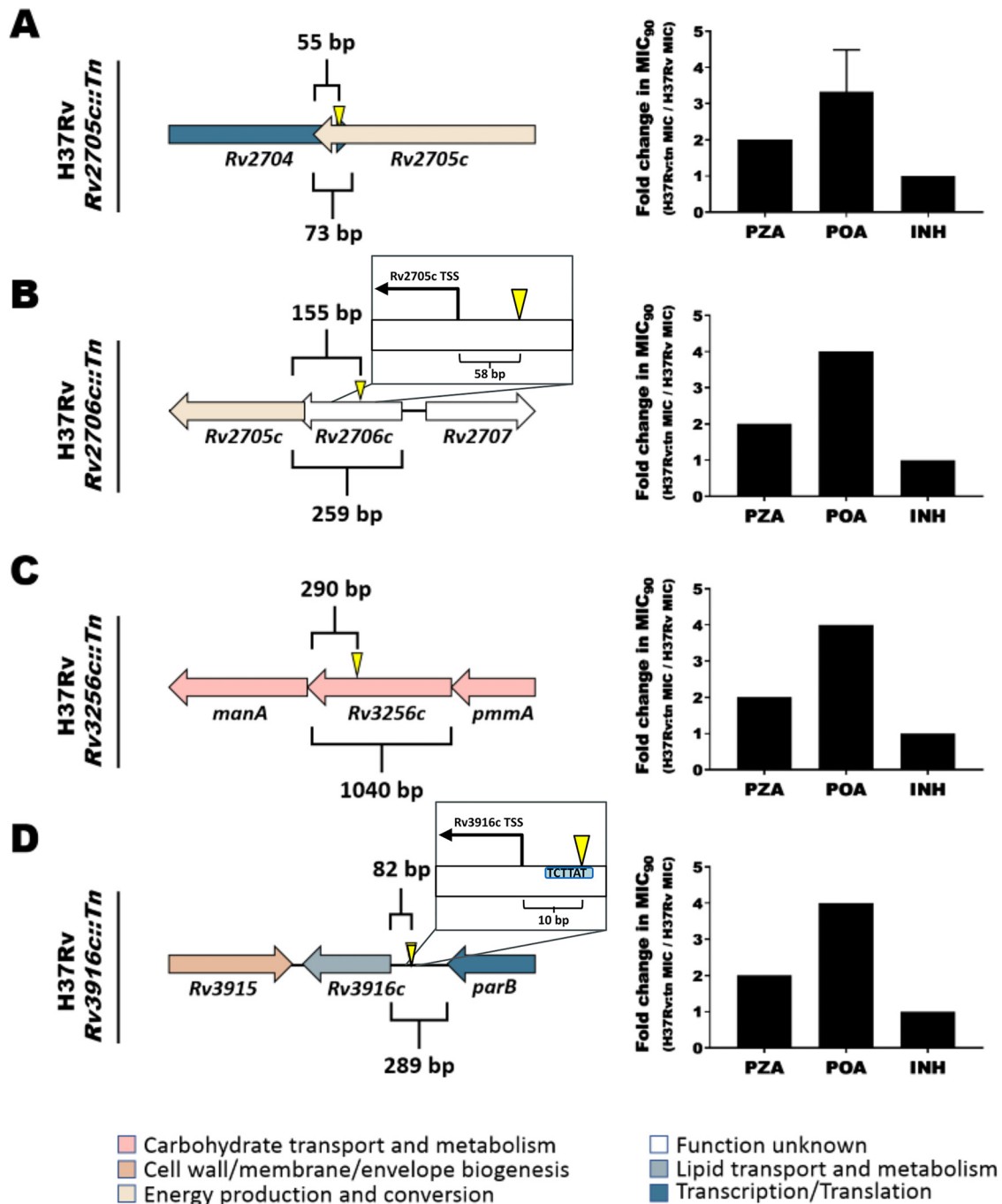

**FIG 7** Annotation of genes involved in pyrazinamide resistance. A library of *M. tuberculosis* H37Rv transposon insertion mutants was used to select for strains that were resistant to POA. The transposon insertion sites were mapped, and the strains were characterized for their susceptibility to PZA, POA, and INH in comparison with wild-type H37Rv (MIC$_{90}$: 50 $\mu$g/ml PZA, 200 $\mu$g/ml POA, 0.0625 $\mu$g/ml INH). (A) H37Rv *Rv2705c*::Tn. (B) H37Rv *Rv2706c*::Tn. (C) H37Rv *Rv3256c*::Tn. (D) H37Rv *Rv3916c*::Tn. Error bar depicts standard deviation across triplicates. Popouts in panels B and D depict the transposon insertion sites relative to experimentally determined transcription start sites (TSS) (103, 126). The insertion in panel D interrupts a TANNNT Pribnow box (blue), destroying the Rv3916c promoter. While the *Rv3916c*::Tn mutant certainly disrupts the Rv3916c promoter, the possibility of nongenic features mediating the PZA-resistant phenotype cannot be dismissed. A recently reported putative noncoding RNA (ncRv13916cA) (127) partially overlaps the transposon insertion site and would ostensibly be interrupted by the transposon insertion. At present, ncRv13916cA has no known functional role. Error bars are shown when there was a deviation in the calculated MIC$_{90}$ across triplicates.

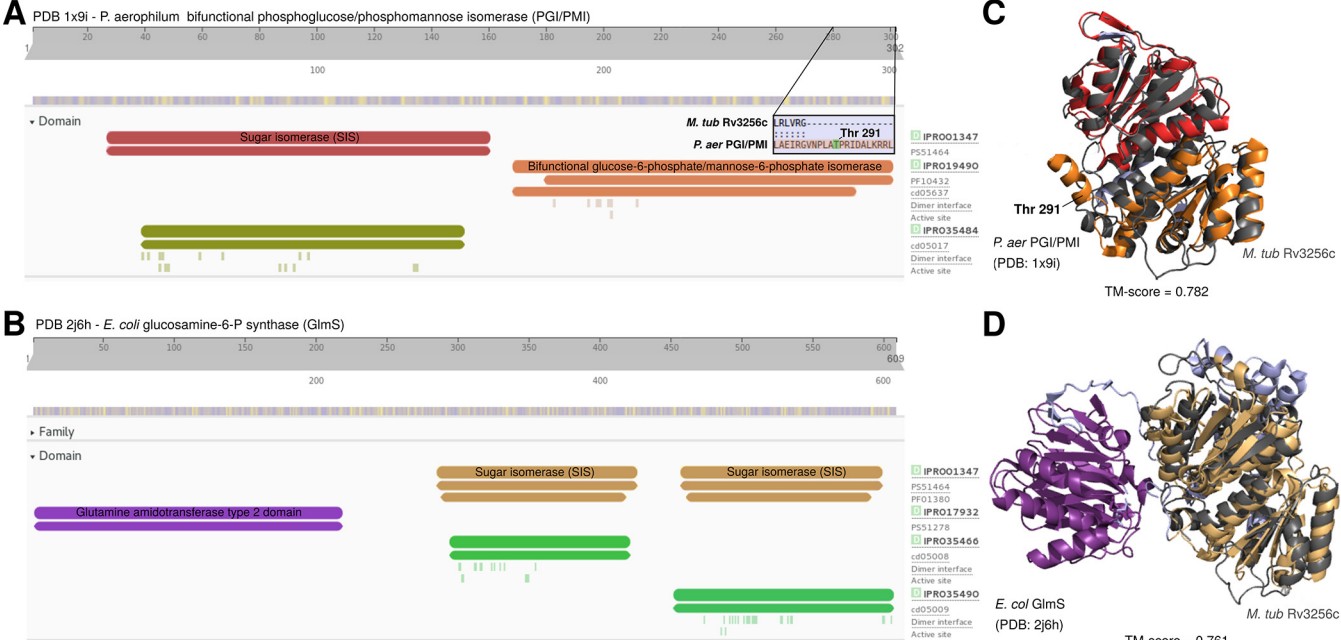

**FIG 8** Structural analysis informs specific functional hypotheses for the basis of PZA resistance in the *Rv3256c*::Tn mutant. (A to D) Structural analysis identifies Rv3256c as a sugar isomerase (SIS) domain-containing protein likely involved in phosphosugar metabolism or its regulation. InterPro functional domains are displayed for the two strongest structural matches of Rv3256c, *Pyrobaculum aerophilum* bifunctional phosphoglucose/phosphomannose isomerase (*P. aer* PGI/PMI) (A) and *Escherichia coli* glucosamine-6-P synthase (*E. col* GlmS) (B). The InterPro domains labeled in panels A and B are mapped onto the three-dimensional (3D) structures of *P. aer* PGI/PMI (C) and *E. col* GlmS (D). Rv3256c (charcoal) modeled protein structure is optimally superposed on each of its matches. Rv3256c is structurally homologous to the SIS domains of *E. col* GlmS and *P. aer* PGI/PMI and exhibits the alpha-beta-alpha sandwich fold of SIS (128). The popout in panel A (**, <5 Å between residues) and labeled residue in panel C show the threonine residue essential for isomerase activity in *P. aer* PGI/PMI (Thr[291]) and other PMI homologs. The Thr[291]-containing region appears to be absent from Rv3256c. Likewise, Rv3256 lacks a glutamine amidotransferase domain homologous to *E. col* GlmS. From this structural evidence, we conclude that Rv3256c is a SIS domain protein putatively involved in phosphosugar metabolism and/or its regulation. Structural images were rendered in PyMOL. Structurally homologous sequence alignments from TM-align (22).

phosphosugar metabolism, though a specific molecular function for Rv3256c remains unclear. Presumably, the role played by Rv3256c in phosphosugar metabolism is disrupted in the Rv3256c::Tn mutant. This disruption may alter composition of acyl-CoA/CoA pools (e.g., through disrupting/promoting acylation of cell wall constituents) or the metabolic and cell wall restructuring response under CoA depletion (93) induced by PZA treatment.

Rv3916c structurally resembles numerous Gcn5-related *N*-acetyltransferase (GNAT) proteins and exhibits structural, topological, and local features characteristic of GNAT enzymes (Fig. 9). Perhaps most tellingly, Rv3916c has a predicted acetyl-CoA binding site consistent with known GNAT enzymes (Fig. 9B and C). While the acyl donor and substrate of Rv3916 remain unclear, the conservation of these structural features involved in acyl-CoA interaction in functionally characterized GNAT superfamily proteins strongly suggests that Rv3916 is a GNAT *N*-acetyltransferase. This putative function implicates the ₚRv3916c::Tn mutant in the CoA depletion model of POA action. Destruction of the Rv3916c promoter would reduce its expression, in turn altering acyl-CoA pool modulation. Structure-function insights gleaned from these structural models inform specific functional hypotheses for these mutants' role in PZA resistance and demonstrate how the provided structural data can enrich the interpretation of large-scale screens and generate specific functional hypotheses.

## DISCUSSION

Functional genome annotation is critical for interpreting the deluge of omics data generated by emerging high-throughput technologies. Here, we devised procedures to systematically curate annotations from published literature and infer putative function through structure-based inference and applied them to annotate the *M. tuberculosis* virulent type strain and primary reference genome, H37Rv. We curated annotations

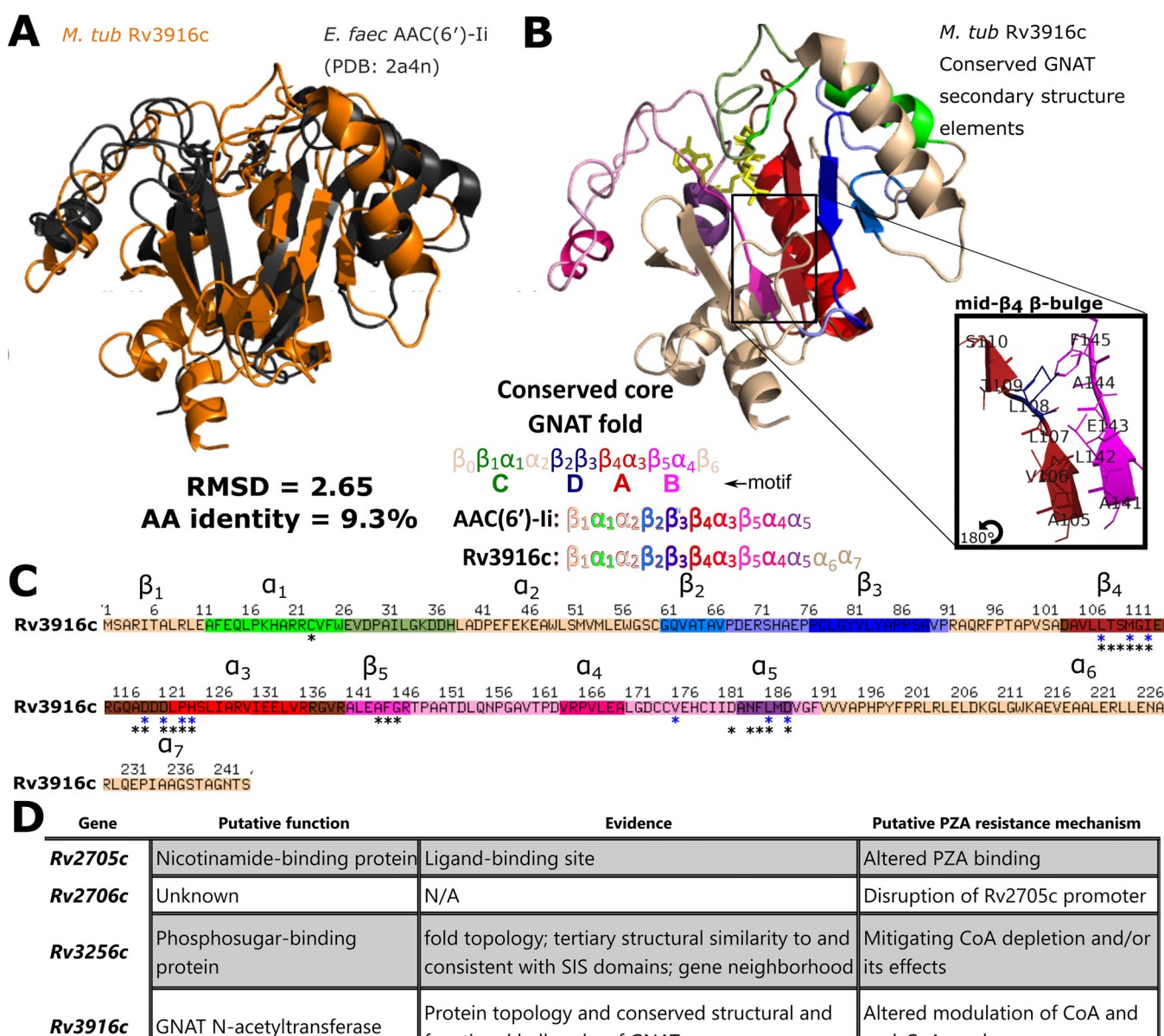

**FIG 9** Structural analysis of transposon mutants refines functional hypotheses for their role in PZA resistance. (A to C) Structural analysis supports Rv3916c as a general control nonrepressible 5 (GCN5)-related *N*-acetyltransferases (GNAT). Rv3916c matches exclusively comprised GNAT enzymes with low sequence similarity, typical among homologous GNAT enzymes (129). (A) Rv3916c (orange) superposed with its closest structural match (PDB ID 2a4n, charcoal), *Enterococcus faecium* aminoglycoside 6′-*N*-acetyltransferase [*E. faec* AAC(6′)-Ii]. RMSD, root mean square deviation. (B) *M. tuberculosis* Rv3916c structural model (top) and secondary structure topology (bottom), colored according to the four core conserved GNAT folds (wheat = poorly conserved secondary structure element) and secondarily (tones of the primary colors) by secondary structure elements in *E. faec* AAC(6′)-Ii (as defined by reference 130) with which Rv3916c structurally aligns. All secondary structure elements of *E. faec* AAC(6′)-Ii are present in Rv3916c, with two gratuitous alpha-helices in its C-terminal arm (which is not well conserved among GNAT enzymes [129]). The distinctive $\beta$-bulge (popout) within the $\beta_4$ strand (red; bulge residues colored blue) characteristic of GNAT enzymes is present in Rv3916c, diverting $\beta_4$ away from $\beta_5$ to create the chasm where the acetyl-coenzyme A (yellow) is predicted to bind. (C) Primary sequence of Rv3916c colored according to the scheme described in panel B. Asterisks mark predicted acetyl-CoA binding residues (black) and residues structurally aligning to known CoA-interacting residues (blue). All known CoA-interacting residues from *E. faec* AAC(6′)-Ii are conserved in Rv3916c, and all predicted acetyl-CoA binding sites coincide with or directly flank demonstrated sites of CoA interaction. The presence of the features in Rv3916c suggests it is a GNAT enzyme. GNAT enzymes catalyze transfer of an acyl moiety from an acyl-CoA to various substrates (129), making Rv3916c a probable acyl-CoA acetyltransferase and implicating pRv3916c::Tn in the CoA pool modulation model of PZA resistance. (D) Summary of functional hypotheses for PZA resistance-conferring transposon mutants. All structural images were rendered in PyMOL. Structurally homologous sequence alignments are based on TM-align (22). N/A, not available.

for hundreds of proteins with published functions lacking from common resources (Table 2), a quarter of which were absent from all five annotation resources examined, highlighting the importance of community-specific manual curation. To complement these manually curated annotations, we built a structural modeling and functional

inference pipeline (Fig. 1 to 4), calibrated it to include confident annotations (Fig. 1), and orthogonally validated it with established remote homology detection methods (Fig. 3). Through structure-function inference we annotated hundreds of genes (Fig. 5), including dozens of potential transport proteins, resistance genes, and virulence factors (see Data Set S1 in the supplemental material).

Elucidating the determinants of *M. tuberculosis* survival under drug pressure and within the context of infection is a chief objective of tuberculosis research. Integrating this updated functional annotation with new (Fig. 7 to 9) and published (Tables 5 and 6) functional screens showed it can aid in understanding the genetic basis of *M. tuberculosis* resistance to drug pressure and infection-like conditions. The structural models provided a rational basis for functional hypotheses of the molecular basis of resistance for four novel PZA-resistant strains with mutations in otherwise unannotated genes (Fig. 7 to 9). In particular, the Rv3256c and Rv3916c mutants implicate CoA homeostasis in PZA resistance (Fig. 8 and 9), linking them to the CoA depletion model of the PZA mode of action described recently for other PZA-resistant (PZA-R) mutants (93). The functional interpretation of transposon insertion sequencing (TnSeq) mutants afforded by this resource informs hypotheses for the mechanistic basis of these mutants' PZA resistance for investigation in future work.

Our systematic approach to manual literature curation has limitations. First is the time and attention from researchers with specialized knowledge required for manual literature curation. To mitigate this limitation in the future, contributions from the TB research community can be submitted and will be incorporated with standardized criteria and structured ontologies (24, 28). A second limitation is the inevitable subjectivity of the curator. We addressed this by requiring that two curators review each paper independently and providing explicit guidelines for what evidence warrants annotation with the degree of confidence connoted systematically by qualifying adjectives.

Other limitations arise from the scope of our annotation. First, we curated functions for only 1,725 of over 4,000 ORFs in *M. tuberculosis*. Products that did not meet our criteria for inclusion may have useful functional characterizations excluded by our approach. Second, we searched only for the locus tag during curation. While most publications include locus tag, some do not, and therefore, some experimental characterizations may remain unannotated. Last, we searched literature back from 2010, as TubercuList updated continuously through March 2013, and we assumed annotations to that point were captured. However, the absence of dozens of characterizations from all resources suggests some findings prior to 2010 may remain unintegrated. Despite these limitations, the numerous genes we curated that were absent from all frequented annotation sources are now centralized in a single updated annotation that is clear in source and confidence level, in a consistent and extendable format.

Multiple factors contribute to error and bias in resolving protein structure and function. These factors fall unevenly across protein classes and families (101), making them challenging to account for. Considering this while designing our structure-based inference pipeline, we favored simple, interpretable inclusion criteria, coupled with downstream quality assurance measures. Future work more focused on customizing inclusion criteria optimized for features of protein structure or function may improve prediction accuracy. Our simplified approach let us circumvent accounting for these biases explicitly, which would require further method development and introduce additional bias if not executed carefully.

Our structural approach to functional inference also has limitations. First, it depends on the input sequence. We took amino acid sequences as provided by TubercuList without accounting for the impact of known, uncorrected sequencing errors (102) or corrections to amino acid sequences proposed by UniProt curators. Furthermore, some genes have multiple translation initiation sites, or isoforms (103), but we considered one sequence per gene. Second, our approach compares global, rather than local, structural similarity and can be challenged by functionally diverse folds (104) and proteins with dynamic active sites (105) or context-specific conformation and activity

(106). Our empirically driven inclusion criteria (Fig. 1) and quality control measures helped to mitigate false-positive annotations (Fig. 3 and 4). In future analysis of structural models, emerging methods that capture functional conservation distributed across primary and tertiary structure may identify functionally informative protein features missed by our approach. Promising approaches include direct coupling analysis (107), statistical coupling analysis (108), Bayesian partitioning with pattern selection (109), and structurally interacting pattern residues' inferred significance (110). Third, proteins from model organisms and humans are overrepresented among crystallized structures on PDB (https://www.rcsb.org/) (111). This adds bias toward inferring function from these proteins. Finally, the structure-based annotations should be interpreted as tentative, since inclusion criteria required similarity implying >50% ("putative") or >75% ("probable") likelihood of being correct. Structure-based annotations should be viewed accordingly, as well-informed hypotheses rather than established truth.

Over half of structure models (871/1,711) were low quality (C-score $< -1.5$) (21) (Data Set S1). Several phenomena may challenge effective modeling of these underannotated genes: (i) no proteins of similar folds have been solved; (ii) the protein is highly disordered (112); (iii) these are multidomain proteins that need to be split into individual domains (14); (iv) sequencing errors; (v) gene coordinate misannotation (102); and (vi) pseudogenization. We suspected reason 3 as a major factor, considering we did not attempt to break up multidomain polypeptides into their constituent domains (14). However, the protein length distributions of proteins of high (greater than $-1.5$) and low (below $-1.5$) C-scores were similar (Text S1 and Fig. S5C), which suggests the presence of multiple domains was not a primary cause of poor models. Each of the other reasons likely contributes to some extent, but reasons 1 and 2 are most troublesome for PE/PPE genes and other protein classes specific to mycobacteria.

I-TASSER failed to produce models for 14 underannotated genes (Data Set S1). Six of these sequences are pseudogenes, and the remaining 8 belong to PPE or PE_PGRS gene families, which are especially prone to sequencing errors and intrinsically hypervariable (113). Although we ascribed putative functions for some PE/PPE genes, the function of most remains unclear. Far fewer PE/PPE proteins (20/166, 12.0%) than non-PE/PPE genes (518/1,559, 33.2%) met inclusion criteria for structure-based annotation ($P = 2.13 \times 10^{-9}$, odds ratio = 0.276; Fisher's exact). This likely owes partly to their intrinsic disorder and partly to their specificity to the *M. tuberculosis* complex (114), which limits the number of homologous structures in PDB with known function, challenging accurate structural modeling and structure-based functional annotation. Moreover, PE/PPE and other effector proteins require precise metabolic contexts or immunological cues, precluding observation of their function *in vitro*. Characterizing function for these genes will require high-throughput biochemical assays and development of techniques that directly assay or precisely reconstruct host microenvironments; formidable challenges, indeed. In the meantime, carefully designed and caveated inferential methods can make valuable surrogates and streamline candidate prioritization for experimental confirmation or more comprehensive *in silico* analysis.

Systematically curated literature and structure-derived annotations are available at https://gitlab.com/LPCDRP/Mtb-H37Rv-annotation. Researchers can file issues to report future published characterizations and submit merge requests to incorporate future functional characterizations. These methods can continue to furnish annotations as functional characterizations are published in the primary literature, structure-function relationships in PDB expand, *M. tuberculosis* gene product functions are determined, and sequence-structure-function prediction tools become more resource efficient.

## MATERIALS AND METHODS

Additional details on methods are provided in Text S1 in the supplemental material.

**Manual curation protocol.** All publications mentioning each of the 1,725 underannotated gene were independently evaluated for annotation-worthy functional characterization by two researchers and quality checked by a third for format and protocol compliance (Text S1 and Fig. S1). Qualifying adjectives were defined by evidence quality and systematically assigned to connote annotation confidence.

Notes relevant to function but insufficient to assign product name were also annotated (Text S1 and Fig. S1).

**Precision benchmarking.** We designed procedures and inclusion criteria to maximize precision (equation 1) and minimize "overannotation" (101): only annotations with 50% or greater precision were included, regardless of source. Whereas other metrics had applicable precision benchmarks (Text S1), EC number and GO terms did not. We assessed how precision of EC number and GO term predictions (equation 2) correlated with similarity metrics. We evaluated which I-TASSER metrics were most predictive of precision (equation 1) through logistic regression (Text S1 and Fig. S2).

$$\text{precision} \quad = \quad \frac{(\text{TP})}{(\text{TP} \ + \ \text{FP})} \qquad (1)$$

where TP is true positive and FP is false positive.

We gauged sequence similarity by amino acid identity (AA%) and structural similarity by Template Modeling score (TM-score). TM-score describes structural similarity from 0 and 1. It represents the average root mean squared deviation across all atoms in the predicted structure with respect to the PDB template model, normalized to remove apparent deviation arising falsely due to local differences (14, 115) (Materials and Methods), allowing proteins of different lengths to be compared (115).

To base inclusion criteria off precision (equation 1), we regressed against sequence and structural similarity metrics: amino acid identity, C-score, TM-score (equation 1, as calculated by Zhang and Skolnick [115]), and the geometric mean of TM-score and AA% ($\mu_{\text{geom}}$) against precision of EC number assignment in a positive-control set of 363 *M. tuberculosis* genes with known function but unknown structure. Because EC numbers and GO terms encode the same fundamental information, although GO terms have many false negatives and were relatively underpowered (Text S1 and Fig. S2), we included both according to the same criteria: $\mu_{\text{geom}}$ values corresponding to 50% ("putative") and 75% ("probable") precision for each tier of specificity (Fig. 1).

$$\text{TM}-\text{score} = (1/L_N) \sum_{i=1}^{L_T} \frac{1}{\left(1+d_i^2+d_0^2\right)} \qquad (2)$$

where $L_N$ is protein length, $L_T$ is the length of the residues aligned to the template, $d_i$ is the distance of the *i*th pair of residues between two structures after an optimal superposition, and $d_0 = 1.24\sqrt[3]{L_N-15}-1.8$, as described by Xu and Zhang, normalizes for protein length (21). TM-score measures the difference between predicted structure and known structure of the putative homolog/analog on PDB. Since we are interested in the similarity between the true (unknown) structure and its putative homolog/analog on PDB, we used an adjusted TM-score, $\text{TM}_{\text{ADJ}}$. $\text{TM}_{\text{ADJ}}$ subtracts from TM-score the expected difference in TM-score between the modeled protein structure and its (unknown) true structure (see equations 2 and 3 and accompanying text in Text S1 for additional discussion of this rationale).

**Training data selection.** We combined 200 randomly selected *M. tuberculosis* protein sequences with known function with 163 manually annotated underannotated genes with "probable" or higher annotation confidence (Data Set S1) to form a set of training genes. We extracted EC numbers and GO terms that were marked as experimentally verified in UniProt (116) from the 363 training genes.

**Annotation inclusion criteria.** Structure-inferred annotations comprised product names, GO terms, EC numbers, CATH topologies, and LBS. We systematically transferred EC numbers and GO terms according to $\mu_{\text{geom}}$ thresholds corresponding to 50% and 75% precision (Fig. 1). LBS and CATH predictions were included according to previous precision benchmarks (21, 117). CATH annotations were retrieved using the REST API of PDB for structure matches surpassing the TM-score corresponding to 50% precision, after correcting TM-score for expected modeling error (Text S1). Underannotated genes with quality models (C-score > −1.5) and a TM-score greater than 0.85 and/or $\mu_{\text{geom}}$, meeting the inclusion criteria for putative EC third digit (0.374, corresponding to a precision > 0.5), and further criteria based on aligned portions and method of UniProt annotation of the PDB template (Text S1). Because transport proteins are more conserved in structure than in sequence relative to globular proteins (12), we weighted structural similarity more heavily than AA% in their inclusion criteria: transport protein annotations were transferred if (i) greater than 90% of the PDB structure implicated in transport aligned with the underannotated gene model and (ii) structural similarity exceeded the threshold for CATH topology transfer. All analyses were implemented in R (118).

**Product naming protocol.** To translate transferred GO terms and EC numbers into product names, we converted GO terms that describe enzymatic activity into EC numbers by searching EXPaSY ENZYME. Product names were converted from EC numbers (including those derived from GO terms) using the ENZYME.dat file from the EXPaSY database (Text S1 and Fig. S3 and S4). GO terms that mapped to multiple EC numbers were merged at the most specific level at which they converged (e.g., 3.2.1.5 and 3.2.2.4 would resolve to 3.2.-.-). When GO terms did not map to an EC number, we translated sufficiently descriptive GO terms into product names (e.g., "DNA binding transcription factor activity" is sufficiently descriptive whereas "pathogenesis" is not). Product names for PDB matches lacking GO or EC annotations were determined manually. Transport proteins were named with lower specificity than their PDB matches (e.g., "transport protein" instead of "Na$^+$/H$^+$ antiporter"), unless (i) all three strongest PDB matches converged on a more specific description and (ii) TM-score exceeded 0.85 for at least one of the three, in which case the name in common between the three strongest matches was transferred.

LBS predictions and the residues predicted to coordinate binding (Data Set S3D) can be interpreted as being at least 60% likely to be true (23), though most have greater confidence.

For proteins annotated only with structure-based functional inferences that had EC number annotation modified by HHpred filtering or had multiple EC numbers corroborated by HHpred, the implicated structural homologs were inspected manually, and spurious or infeasible annotations were pruned. Reasons for pruning EC numbers include cases where one of the implied catalytic functions was exceedingly unlikely (such as eukaryotic proteins with bacterial homologs that had evolved distinct, nonoverlapping functions) or there was a clear reason for a false positive (such as structural alignment with a multifunctional protein to only one of the functional domains). Additional annotation specificity was added in rare cases, where HHpred results strongly corroborated evidence from structural alignment that alone did not meet inclusion criteria for specific annotation. Rv3433c exemplifies such cases. It was annotated with EC 4.2.1.- and EC 5.-.-.-, and both were corroborated by HHpred. Upon inspection, the annotators noted that the top hits from structural alignment and HHpred were a mixture of EC 5.1.99.6 and EC 4.2.1.136 proteins and bifunctional proteins encoding both catalytic functions. The portions aligning to the respective EC functions were mutually exclusive, and Rv3433c was of similar length as characterized bifunctional enzymes including both functions. In this case, EC numbers were updated to full specificity and the product name was changed from "putative hydro-lyase/putative isomerase" to "putative bifunctional NAD(P)H-hydrate repair enzyme."

**Comparison with other databases.** To assess the novelty of manual product annotations, we compared our annotation for each underannotated gene with the corresponding entry on UniProt (116), Mtb Network Portal (9) (which included annotations from TBDB [5]), PATRIC (6), RefSeq (36), BioCyc (119), and KEGG (120). Comparisons were performed programmatically where possible and systematically otherwise. Annotations were retrieved on the following dates: 17 May 2017 for RefSeq (36) (https://www.ncbi.nlm.nih.gov/refseq/), PATRIC (6) (https://www.patricbrc.org/), and Mtb Network Portal (9) (http://networks.systemsbiology.net/mtb/) and 23 June 2017 for KEGG (120) (https://www.kegg.jp/kegg/genome/pathogen.html) and UniProt (116) (https://www.uniprot.org/uniprot/).

**Enzyme Commission number assignment.** We assigned EC numbers to underannotated genes with experimentally verified enzymatic activity using that assigned by the source article's author when compliant with IUBMB standards. Otherwise, we manually assigned one using the official IUBMB database (25).

**HHpred filtering of structure-inferred functional annotations.** All proteins with functions assigned solely by structural inference were run through HHpred, searching against the PDB70 and ECOD databases, limiting maximum number of hits to 1,000, and using default parameters for the searches. All HHpred results were filtered, and only hits where Prob was >0.95 were retained for downstream analysis. Each function assigned to a protein was evaluated separately (e.g., a bifunctional protein could have one function culled and the second function retained). Annotations were evaluated differently depending on whether they had a corresponding EC number or not. All annotations with an EC number assigned were evaluated programmatically and retained to the degree of EC specificity matched by the HHpred hit(s). Annotations without corresponding EC numbers were evaluated manually, independently, by two curators. Each curator screened all HHpred hits and evaluated whether HHpred hits supported the assigned function wholly, entirely, or not at all. In cases where function was partially supported, each curator submitted a suggested product name change. After evaluating all proteins, the curators reconciled any disparate assignments. Functions entirely uncorroborated by HHpred that passed the Ramachandran plot analysis filtering step were subsequently evaluated to determine whether the structural similarity used to infer function had substantial evidence warranting a transferred annotation. Original structural inferences were either discarded, retained, or modified at the discretion of the curators. For an annotation to be accepted, curators verified that model proteins were not threaded on low-complexity proteins, checking whether regions underlying the function of the structurally solved protein structurally aligned to the protein model being annotated and for conservation of any known residues or structural motifs essential for function.

**Ramachandran plot analysis.** To evaluate structure model quality, we computed the fraction of the residues in "most favored" regions, "additionally allowed" regions, "generously allowed" regions, and "disallowed regions," via the PROCHECK server (121). The .pdb file containing the atomic coordinates of each model protein structure of interest was uploaded to PROCHECK, and proportions of residues in each regional favorability classification were extracted from the "results summary" file and collated into tabular format for further analysis. To determine a threshold for including structures not corroborated by HHpred on the basis of quality protein structure, we assessed the distributions of "most favored" region residues between proteins wholly uncorroborated by HHpred and those fully corroborated by HHpred with maximally specific EC numbers (Fig. 3).

**Structural model visualization and annotation.** Protein structure models (.pdb files) from I-TASSER and solved protein structures form the Protein Data Bank were visualized and annotated with PyMOL (https://pymol.org/).

**Comparison with annotations from recent functional screens.** Tables S2 and S4 from a recently published transposon mutant functional screen (83) were downloaded, and the intersection of their locus tags and the underannotated gene set of this study went on for further analysis. Genes with functions annotated by Bellerose et al. (83) were filtered out. PE/PPE genes and transcriptional regulatory proteins were also excluded, as the novelty comparison was determined according to product name, which typically remains generic for these two classes of proteins even upon updated functional information.

**Bacterial strains and growth media.** *M. tuberculosis* strain H37Rv was a gift from W. R. Jacobs, Jr., of the Albert Einstein College of Medicine. Strains were grown in Middlebrook 7H9 medium (Difco)

supplemented with 10% (vol/vol) oleic acid-albumin-dextrose-catalase (OADC) (Difco), 0.2% (vol/vol) glycerol, and 0.05% (vol/vol) tyloxapol.

**Characterization of POA-resistant strains.** Strain H37Rv was mutagenized with the *mariner*-based transposon (122, 123). Approximately $10^5$ independent transposon-mutagenized bacilli were plated on Middlebrook 7H10 medium supplemented with 10% (vol/vol) OADC (Difco) and 0.2% (vol/vol) glycerol with 50 $\mu$g/ml of pyrazinoic acid (POA) (Sigma). Resistant mutants were selected from the plates containing 50 $\mu$g/ml POA. The initial isolates were plated on 7H10 medium supplemented with 10% (vol/vol) OADC (Difco) and 0.2% (vol/vol) glycerol containing either 400 $\mu$g/ml POA or no drug after their initial isolation to confirm their POA resistance prior to the more detailed drug susceptibility testing (123, 124). Transposon insertion sites were identified as previously described.

The antimicrobial drug susceptibility was determined by assessing the minimum concentration of drug that was required to inhibit 90% of growth ($MIC_{90}$) relative to a no-drug control. Growth was assessed by measuring optical density at 600 nm ($OD_{600}$) of cultures after 14 days of incubation at 37°C. Drug susceptibility testing for PZA and POA was carried out in 7H9 broth supplemented with OADC, glycerol, and tyloxapol (pH 5.8) as indicated above. INH $MIC_{90}$ determinations were carried out in medium with the same composition at pH 6.8.

**Data availability.** We provide final annotations in common machine (GFF3) and human (Data Set S1) readable formats, including EC numbers, GO terms, CATH topologies, and product name annotations. Annotations in the GFF3 are defined by our inclusion criteria. PDB templates with structures similar to yet below our criteria are provided in Data Set S3B (top 3 PDB templates for each underannotated gene) and Data Set S3A (all matches where $TM_{ADJ} > 0.52$ [equation 2, Text S1] and/or $\mu_{geom} >$ EC3 [putative] threshold). I-TASSER results and model protein structures for 1,711 underannotated genes are freely accessible at https://tuberculosis.sdsu.edu/structures/H37Rv/ including functional predictions by COFACTOR, predicted ligand binding sites, local secondary structure confidence (B-factor), and other quality and similarity metrics.

## SUPPLEMENTAL MATERIAL

Supplemental material is available online only.
**TEXT S1**, DOCX file, 0.1 MB.
**FIG S1**, TIF file, 0.5 MB.
**FIG S2**, TIF file, 1.1 MB.
**FIG S3**, TIF file, 1.6 MB.
**FIG S4**, TIF file, 0.6 MB.
**FIG S5**, TIF file, 0.5 MB.
**DATA SET S1**, XLSX file, 0.6 MB.
**DATA SET S2**, XLSX file, 0.1 MB.
**DATA SET S3**, XLSX file, 4.7 MB.
**DATA SET S4**, TXT file, 1.7 MB.

## ACKNOWLEDGMENTS

We extend our gratitude to Vikram Alva for running a large set of proteins through HHpred in batch mode on our behalf. We thank Sarah Radecke, Derek Conkle-Gutierrez, and Matthew Onorato for their extensive review of the manuscript and helpful comments. We also acknowledge Yusuke Minato for his contributions to PZA experiments.

This work was funded by grants from National Institute of Allergy and Infectious Diseases (NIAID grant no. R01AI105185 to F.V. and R01AI123146 to A.D.B.). S.J.M., A.M.Z., D.G., A.E., N.K., C.R., N.D., C.K.C., and F.V. were supported by R01AI105185. S.J.M. was also supported by scholarships from a National Science Foundation DUE training grant to F.V. (0966391). A.D.B. and N.A.D. were funded by R01AI123146. N.A.D. was also supported by NHLBI (HL007741). The funding bodies had no role in the design of the study or in collection, analysis, and interpretation of data or in writing the manuscript.

Author contributions: S.J.M., A.E., and F.V. conceptualized the overarching research goals and aims; S.J.M., A.E., D.G., A.M.Z., N.K., and C.K.C. developed the manual curation protocol; S.J.M., D.G., A.M.Z., N.K., and C.K.C. manually curated annotations from literature and independently validated annotations; S.J.M., A.E., D.G., A.M.Z., N.K., and C.R. developed and tested code for data processing and analysis; A.E. and S.J.M. developed the structural comparison workflow; S.J.M., A.E., and N.D. developed and executed the structural QC workflow; N.A.D. and A.D.B. conceptualized the PZA resistance experiments; N.A.D. isolated and characterized the PZA-resistant mutants; S.J.M., A.E., D.G., A.M.Z., N.A.D., and N.D. prepared the manuscript; S.J.M., D.G., A.M.Z., and N.A.D. prepared figures and tables; S.J.M. and F.V. managed the project.

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
