## [Reviewer comments · mSystems]

Structure-aware *M. tuberculosis* functional annotation uncloaks resistance, metabolic, and virulence genes

Samuel Modlin, Afif Elghraoui, Deepika Gunasekaran, Alyssa Zlotnicki, Nicholas Dillon, Nermeeta Dhillon, Norman Kuo, Cassidy Robinhold, Carmela Chan, Anthony Baughn, and Faramarz Valafar

Corresponding Author(s): Faramarz Valafar, San Diego State University

Review Timeline:

Submission Date:	June 24, 2021
Editorial Decision:	August 9, 2021
Revision Received:	September 30, 2021
Accepted:	October 4, 2021

Editor: John McGrath

Reviewer(s): Disclosure of reviewer identity is with reference to reviewer comments included in decision letter(s). The following individuals involved in review of your submission have agreed to reveal their identity: Timofey Skvortsov (Reviewer #1); Dhvani Desai (Reviewer #3)

Transaction Report:

DOI: <https://doi.org/10.1128/mSystems.00673-21>

August 9, 2021

Dr. Faramarz Valafar
San Diego State University
Laboratory for Pathogenesis of Clinical Drug Resistance and Persistence
6367 Alvarado Court
Suit 206
San Diego, California 92120

Re: mSystems00673-21 (Structure-aware *M. tuberculosis* functional annotation uncloaks resistance, metabolic, and virulence genes)

Dear Dr. Faramarz Valafar:

Thank you for submitting your manuscript to mSystems. We have completed our review and I am pleased to inform you that, in principle, we expect to accept it for publication in mSystems. However, acceptance will not be final until you have adequately addressed the reviewer comments.

Preparing Revision Guidelines

For complete guidelines on revision requirements for your article type, please see the journal Article Types requirement at <https://journals.asm.org/journal/mSystems/article-types>. **Submissions of a paper that does not conform to mSystems guidelines will delay acceptance of your manuscript.**

Sincerely,

John McGrath

Editor, mSystems

Journals Department
Reviewer comments:

Reviewer #1 (Comments for the Author):

This is a resubmission of the manuscript by Modlin et al. that I previously reviewed. The main point of my initial criticism was that some of the protein structure predictions were incorrect due to a number of factors, including the subpar quality of certain templates used by I-TASSER for initial modelling and the lack of the protein model validation steps. As the assignment of functions is based on the transfer of annotations from the proteins in the PDB showing structural similarity to the newly generated protein models, incorrect assumptions about the nature and functional role of some proteins (e.g. Rv3507 and Rv0193c) were made. I am pleased to see that the authors have addressed this issue by introducing additional checks and critically evaluating the newly obtained results, as well as being more reserved and careful with their speculations in this version of the manuscript. I believe that this labour-intensive effort resulted in a substantial improvement of the paper overall. I can also confirm that I was able to access the website with I-TASSER protein models (<https://tuberculosis.sdsu.edu/structures/H37Rv/>) this time.

After carefully reading the manuscript, I would like to make the following comments and suggestions:

Main text:

Line 246-94, Table 3. This is one of the most interesting results for me, as I personally expected HHpred results to be generally more reliable compared to the ones produced by I-TASSER + TM-align (this is based purely on my personal experience with both packages and not on any rigorous testing), so I consider the identification of good quality, manually verified models whose functional annotations were not concordant with the ones derived from HHpred results to be a somewhat unusual result.

I have submitted three protein sequences (taken from mycobrowser.epfl.ch) for a search using HHpred (with the same parameters as you specify in the Materials and Methods section (DBs: PDB70 and ECOD, max target hits: 1000, everything else - default parameters)

Rv2036 - <https://toolkit.tuebingen.mpg.de/jobs/4582819>
Top hit - 2NSF_A

Rv1632c - <https://toolkit.tuebingen.mpg.de/jobs/1991452>
Top hit - 2P12_A

Rv1775 - <https://toolkit.tuebingen.mpg.de/jobs/6529826>
Top hit - 3HWP_A

These are the same results as you report for Rv2036, Rv1775 and Rv1632c top structurally similar proteins from PDB (Table 3 - PDB ID), so it is quite surprising why in your case HHpred was unable to detect them. You might want to double-check the remaining 6 proteins from Table 3 to see if all HHpred results are in agreement with your functional annotations this time.

Line 80-82

"One alternative approach is identifying protein homologs and analogs through shared structure, which offers considerable advantages. First, it removes bias toward a priori assumptions by not limiting search space to evolutionarily close relatives"
Line 246-248

"Since HHpred is designed to detect homology between proteins(43) (but not necessarily analogy), there may be genuine functions inferred by our structural similarity pipeline that HHpred did not corroborate."

HHpred uses secondary structure similarity data to improve the sensitivity of detection of similarity between two proteins, so it can identify structurally analogous proteins as a significant hit (although the developers of HHpred consider it a drawback, rather than a feature (<https://www.ncbi.nlm.nih.gov/pmc/articles/PMC1160169/>): "Compared to methods that rely on pairwise comparison of simple sequence profiles, HHsearch gains sensitivity by using position-specific gap penalties. If the default setting 'Score secondary structure' is active, a score for the secondary structure similarity is added to the total score. This increases the sensitivity for homologous proteins considerably (28). As a possible drawback, it may lead to marginally significant scores for structurally analogous, but non-homologous proteins"). At the same time, as I-TASSER is not a de novo protein structure prediction tool, it relies on the generation of deep MSAs (similar to the MSA required for the HHpred initial HMM modelling), so certain, although very small degree of sequence similarity between the protein of interest and PDB templates used for generation of I-TASSER initial models is expected. While it is true that TM-align is able to detect structure analogy without any underlying homology (which HHpred cannot do), I personally have rarely, if ever, seen the situation when TM-align would correctly identify the best structural analogue to the newly generated model that wasn't amongst the starting modelling templates and/or didn't have at least distant homology to the I-TASSER protein model. Nevertheless, I believe the direct comparison between HHpred and I-TASSER + TM-align results is well beyond the scope of this paper.

Line 414-464 and Fig 7-9.

Thank you for your explanation in the Response to Reviewers as to why the four hits of interest were not included in the master annotation file.

I am still not entirely convinced that the information obtained from structural models has significantly improved our understanding of the mode of action of the genes identified by mariner transposon mutagenesis as implicated in PZA resistance, at least when it comes to Rv2705c and Rv2706c:

- Rv2705c (Line 432-433, Fig 9D). Rv2705c where does the information that it contains a putative nicotinamide binding domain come from (apologies if I missed it, but I have checked both the text of the paper and the Datasets 1 and 3)?
- Rv2706c (Line 434-435, Fig 9D). No structural information was used (or required) to make the prediction that Rv2706c might be involved in the disruption of Rv2705c, the assumption was made solely based on the relative positions of genes.

Fig 7A - judging by the figure, the 3' ends of Rv2704 and Rv2705c seem to overlap and the transposon insertion site is located right in that overlap between the genes. You don't discuss the possibility that Rv2704 might be involved in PZA resistance too or that it is the disruption of Rv2704 rather than of Rv2705 gene sequence causes PZA resistance.

Rv3916 (Fig 7C). I agree that the transposon insertion appears to affect the Rv3916c promoter, but could you please also check if any putative ncRNA genes are present in the intergenic region between Rv3916c and parB? Non-coding RNAs are abundant in mycobacteria and might be potentially involved in the regulation of antibiotic resistance as well.

Minor comments:

1. Please review your References section again - the very first reference is still lacking the necessary information (year, journal name, volume/issue/page numbers) and I believe the same is true for Ref 30 and a number of others:
 1. Poux S, Arighi CN, Magrane M, Bateman A, Wei C-H, Lu Z, Boutet E, Bye-A-Jee H, Famiglietti ML, Roechert B. On expert curation and scalability: UniProtKB/ Swiss-Prot as a case study.
2. The term 'mannotation' is still used in the Supplementary Materials (e.g. Fig S1, Dataset 2); I would suggest just adding a note describing your short-hand notation somewhere in the manuscript (just before the legends for Supplementary data, for example).
3. Fig. S2B - match/mismatch colours mixed up? (I think match should be teal and mismatch - red?)
4. Line 162-163: Rv1430 is in UniProt (EC 3.1.1.-) and has been present in Uniprot since version 45 of the gene record: <https://www.uniprot.org/uniprot/L7N697>. I presume you had conducted your literature analysis before the UniProt entry was updated to include the EC code, so maybe you can add the dates when the data was retrieved from UniProt and other databases you used in the Materials and Methods section?
5. Supplementary text, p. 9, first paragraph. I believe that an unrelated fragment of text was copy-pasted into the second sentence of the paragraph ("Many mutations that altered bacterial clearance...")
6. Supplementary text, p. 12, final paragraph. It should be Rv1191, not Rv1191c. Could you also add a short explanation why you believe it should be classified as a cathepsin (what protein did you transfer this annotation from)?

Reviewer #3 (Comments for the Author):

In this manuscript, Modlin et al., attempt to tackle the problem of assigning functions to ~1700 hypothetical and/or under-annotated genes in the Mycobacterium tuberculosis H37Rv (Mtb) genome. Rapid and accurate annotation of microbial genomes is indeed a very critical and under appreciated part of microbial ecophysiology. This step is especially crucial for pathogenic organisms such as Mtb where accurate functional annotation of these hypothetical proteins could unravel mechanisms which could act as drug targets.

The authors employed a two-pronged strategy to define a set of these unannotated or under-annotated genes and to then provide possible functions for many of these genes. First, they undertook a large-scale manual curation of literature to assign functions (including EC numbers for enzymatic functions) to ~575 genes. These annotations were compared across the known annotation resources for Mtb such as UniProt, the Mtb Network Portal, PATRIC, and RefSeq. The second approach used by the authors involved a pipeline for structural modelling of proteins to infer function based on shared 3D structure. Combining these two approaches the authors were able to provide annotations for 623 such under-annotated proteins.

The manuscript is generally well written (given the fact that it has already been reviewed before) and easy to follow. The methods employed are sound and extensive, and the results are described and discussed adequately.

Homology based protocols are limited by the lack of sufficient sequence similarity and can be complemented by context based methods. This is one aspect that the authors have not discussed in detail.

The central idea of expanding the functional repertoire of Mtb is not novel and even structure based alignment has been attempted before (PMID: 31113361, PMID: 33076816). As such, the combined approach of extensive literature survey and the large-scale structural modelling is unique. How do the novel predictions of function in this manuscript match up with other similar

predictions?

Summary:

In this manuscript, the authors attempt to quantify the contributions of picocyanobacteria (specifically the genera *Prochlorococcus* and *Synechococcus*) to the total primary production in the North Pacific Ocean. The authors employed flow cytometry to estimate the abundance of the picocyanobacteria and stable isotope mass spectrometry to estimate the Carbon and Nitrogen uptake rates. They have reported that the abundance of *Prochlorococcus* and *Synechococcus* was differentially distributed with the former dominant in the tropical Pacific and the latter more abundant in the subtropical and temperate Pacific. They also report that these picocyanobacteria contribute ~45% of the total primary production in the tropical Pacific and ~70% of total in the subtropical and temperate Pacific.

Strengths:

The manuscript is generally well written and employs sound scientific techniques. The rationale for the study is introduced properly and the results adequately described and discussed.

Major Criticism:

The Fouilland et al paper that the authors cite for their method of using Cycloheximide as a protein synthesis inhibitor specific for Eukaryotes, suggests using Cycloheximide in conjunction with Streptomycin (to specifically inhibit Bacteria) to get relative contributions of heterotrophic bacteria and phytoplankton to the NO₃⁻, NH₄⁺ and urea uptake rates. Could the authors explain their rationale to omit using Streptomycin (or other Bacteria specific inhibitors) in this study? Heterotrophic bacteria have also been shown to contribute substantially to both Carbon fixation (dark) as well as nitrogen fixation. It might be useful to know the estimates of heterotrophic contributions to the overall productivity in the Pacific ocean.

Minor Criticism:

Line 41: Typo; change to “among dominant water masses”

Line 43: Typo; change to “long-term research”

Line 51: Typo; change to “physico-chemical”

Lines 154-155: The fact that temperature and salinity measured for the SP zone was only measured at one station should be mentioned clearly here.

Lines 168-169: Again, the mean euphotic depth for SP is coming only from one station (and hence is not technically a mean value across the region). The authors should be careful in phrasing this clearly here.

Lines 196-202: The abundance values (cells m⁻²) are not formatted scientifically. The exponential values should be superscripted.

Line 278: Typo; change to “physico-chemical”

Lines 279-281: Grammatically incomplete sentence; Is it supposed to be in continuation with the previous sentence?

Line 287: “typical of temperature oligotrophic waters” Is it supposed to mean “high temperature oligotrophic waters”?

Lines 301-303: Grammatically incorrect; Perhaps the authors meant to write “Based on the hourly carbon uptake rates by total phytoplankton **which** were estimated in this study”

Line 313: The ± sign is missing in the Nitrate and Ammonium uptake rates for the TP region.

Dear Editor,

Please find point-by-point responses to issues raised by the reviewers below, in blue

Reviewer #1 (Comments for the Author):

This is a resubmission of the manuscript by Modlin et al. that I previously reviewed. The main point of my initial criticism was that some of the protein structure predictions were incorrect due to a number of factors, including the subpar quality of certain templates used by I-TASSER for initial modelling and the lack of the protein model validation steps. As the assignment of functions is based on the transfer of annotations from the proteins in the PDB showing structural similarity to the newly generated protein models, incorrect assumptions about the nature and functional role of some proteins (e.g. Rv3507 and Rv0193c) were made. I am pleased to see that the authors have addressed this issue by introducing additional checks and critically evaluating the newly obtained results, as well as being more reserved and careful with their speculations in this version of the manuscript. I believe that this labour-intensive effort resulted in a substantial improvement of the paper overall.

I can also confirm that I was able to access the website with I-TASSER protein models (<https://tuberculosis.sdsu.edu/structures/H37Rv/>) this time.

After carefully reading the manuscript, I would like to make the following comments and suggestions:

Main text:

Line 246-94, Table 3. This is one of the most interesting results for me, as I personally expected HHpred results to be generally more reliable compared to the ones produced by I-TASSER + TM-align (this is based purely on my personal experience with both packages and not on any rigorous testing), so I consider the identification of good quality, manually verified models whose functional annotations were not concordant with the ones derived from HHpred results to be a somewhat unusual result.

I have submitted three protein sequences (taken from mycobrowser.epfl.ch) for a search using HHPred (with the same parameters as you specify in the Materials and Methods section (DBs: PDB70 and ECOD, max target hits: 1000, everything else - default parameters)

Rv2036 - <https://toolkit.tuebingen.mpg.de/jobs/4582819>

Top hit - 2NSF_A

Rv1632c - <https://toolkit.tuebingen.mpg.de/jobs/1991452>

Top hit - 2P12_A

Rv1775 - <https://toolkit.tuebingen.mpg.de/jobs/6529826>

Top hit - 3HWP_A

These are the same results as you report for Rv2036, Rv1775 and Rv1632c top structurally similar proteins from PDB (Table 3 - PDB ID), so it is quite surprising why in your case HHPred was unable to detect them. You might want to double-check the remaining 6 proteins from Table 3 to see if all HHPred results are in agreement with your functional annotations this time.

We thank the reviewer for their detailed and constructive feedback. The job links you supplied returned 404 errors (presumably there was an expiration on how long the job results were kept). However, we re-examined the data from the batch HHPred evaluated in our analysis for these proteins. It is correct that, for the examples you tried, as well as five of the six the remaining proteins in Table 3, HHPred results include a strong hit for the same PDB entry as was the top hit for I-TASSER. However, they were not considered as functionally "HHPred-corroborated" in our analysis because they had neither an EC number nor a functionally informative name output by HHPred (which is how we screened through the thousands of HHPred hits). More importantly, many of these proteins matched numerous hits with similarly high confidence in the HHPred output, making the task of manually sifting through all PDB IDs and referencing the associated papers for potential functional information prohibitively laborious. Without manual curation from the PDB entry, the functional information would not be pulled in from HHPred's output.

This contrasts with the automated retrieval from PDB entries form a much more sharply defined set of fewer strong hits per gene--and, where needed, subsequent manual refinement of function from the single or small set of very strong hits from I-TASSER. We have added language to table 3 clarifying this.

Line 80-82

"One alternative approach is identifying protein homologs and analogs through shared structure, which offers considerable advantages. First, it removes bias toward a priori assumptions by not limiting search space to evolutionarily close relatives"

Line 246-248

"Since HHpred is designed to detect homology between proteins(43) (but not necessarily analogy), there may be genuine functions inferred by our structural similarity pipeline that HHpred did not corroborate."

HHpred uses secondary structure similarity data to improve the sensitivity of detection of similarity between two proteins, so it can identify structurally analogous proteins as a significant hit (although the developers of HHpred consider it a drawback, rather than a feature

(<https://www.ncbi.nlm.nih.gov/pmc/articles/PMC1160169/>): "Compared to methods that rely on pairwise comparison of simple sequence profiles, HHsearch gains sensitivity by using position-specific gap penalties. If the default setting 'Score secondary structure' is active, a score for the secondary structure similarity is added to the total score. This increases the sensitivity for homologous proteins considerably (28). As a possible drawback, it may lead to marginally significant scores for structurally analogous, but non-homologous proteins"). At the same time, as I-TASSER is not a de novo protein structure prediction tool, it relies on the generation of deep MSAs (similar to the MSA required for the HHpred initial HMM modelling), so certain, although very small degree of sequence similarity between the protein of interest and PDB templates used for generation of I-TASSER initial models is expected.

While it is true that TM-align is able to detect structure analogy without any underlying homology (which HHPred cannot do), I personally have rarely, if ever, seen the situation when TM-align would correctly identify the best structural analogue to the newly generated model that wasn't amongst the starting modelling templates and/or didn't have at least distant homology to the I-TASSER protein model.

Nevertheless, I believe the direct comparison between HHPred and I-TASSER + TM-align results is well beyond the scope of this paper.

We thank the reviewer for highlighting this issue and for pointing us to the relevant resources to align our explanation of differences between HHPred and I-TASSER with that explained in the literature. We have now adjusted the text accordingly to focus less on analogy vs. homology, but rather on homology detectable by sequence similarity versus structural analogy or distant homology (lines 84-94).

We have also slightly amended the text you noted (lines 252-254) to acknowledge that HHPred does sometimes identify structural analogs, citing the work you referred us to.

To the reviewer's point regarding TM-align rarely identifying a structural analog that lacked (at least distant) homology to one of the starting templates – our intention is not necessarily to claim this. It is merely to say that some functional predictions inferred by our structural homology pipeline did not show in the HHPred results, yet their functions were consistent with specific features and functional moieties experimentally demonstrated for the protein family with apparent homology/analogy, and the transferred functional annotations. We agree that direct comparison between HHPred and I-TASSER + TM-align results is beyond the scope of this paper. We do however very much appreciate the feedback and constructive criticism provided in this vein throughout the review process.

Line 414-464 and Fig 7-9.

Thank you for your explanation in the Response to Reviewers as to why the four hits of interest were not included in the master annotation file.

I am still not entirely convinced that the information obtained from structural models has significantly improved our understanding of the mode of action of the genes identified by mariner transposon mutagenesis as implicated in PZA resistance, at least when it comes to Rv2705c and Rv2706c:

We thank the reviewer for this feedback. For Rv2705 and Rv2706, we agree that the information obtained in the models does not significantly improve our understanding of their role in PZA/POA resistance, but rather provides potential hypotheses (lines 435-444). It is for this reason that we dedicated much more space to Rv3916c and Rv3256c in the text and figures. We were sure to clearly state that it remains difficult to confidently ascribe function to Rv2706c (Line 439-440). Because this exercise was prospective in nature, we included all such genes in the analysis in order to not misrepresent how invariably the structural information provides significant insight.

- Rv2705c (Line 432-433, Fig 9D). Rv2705c where does the information that it contains a putative

nicotinamide binding domain come from (apologies if I missed it, but I have checked both the text of the paper and the Datasets 1 and 3)?

The putative nicotinamide binding site comes from the Ligand-binding site prediction made by COACH in the I-TASSER suite, which is publicly available (as stated throughout the manuscript):

See <https://tuberculosis.sdsu.edu/structures/H37Rv/Rv2705c/>

- Rv2706c (Line 434-435, Fig 9D). No structural information was used (or required) to make the prediction that Rv2706c might be involved in the disruption of Rv2705c, the assumption was made solely based on the relative positions of genes.

It is true that the potential of the Rv2706- disrupting mutant to confer PZA resistance by affecting Rv2705c does not require structural information. However, we do not state that it does. We state that structural information is used to generate rational function hypotheses for the molecular basis of PZA resistance of four transposon mutants, which remains the case: The structure of Rv2705c informed its predicted Nicotinamide binding site.

Fig 7A - judging by the figure, the 3' ends of Rv2704 and Rv2705c seem to overlap and the transposon insertion site is located right in that overlap between the genes. You don't discuss the possibility that Rv2704 might be involved in PZA resistance too or that it is the disruption of Rv2704 rather than of Rv2705 gene sequence causes PZA resistance.

We have added a brief discussion of this consideration to discussion of Rv2705's functional interpretation (lines 441-444)

Rv3916 (Fig 7C). I agree that the transposon insertion appears to affect the Rv3916c promoter, but could you please also check if any putative ncRNA genes are present in the intergenic region between Rv3916c and parB? Non-coding RNAs are abundant in mycobacteria and might be potentially involved in the regulation of antibiotic resistance as well.

We agree that this is an important exercise. We have checked several databases and sRNA and ncRNA finding analyses (<https://www.pnas.org/content/115/25/6464/tab-figures-data> , <https://doi.org/10.1093/abbs/gmw037> , <https://doi.org/10.1186/s12864-020-6573-5>) for potential overlap with the region interrupted in the *Rv3916c::Tn* mutant. One of the recent publications in fact does report a putative ncRNA whose coordinates overlap the Transposon insertion site. We have now added a couple sentences (lines 1,202-1,205) in the manuscript discussing that the possibility of the ncRNA mediating the observed PZA-resistance cannot be dismissed, and refer to the paper that reported it.

Minor comments:

1. Please review your References section again - the very first reference is still lacking the necessary information (year, journal name, volume/issue/page numbers) and I believe the same is true for Ref 30 and a number of others:

1. Poux S, Arighi CN, Magrane M, Bateman A, Wei C-H, Lu Z, Boutet E, Bye-A-Jee H, Famiglietti ML, Roechert B. On expert curation and scalability: UniProtKB/ Swiss-Prot as a case study.

We thank the reviewer for highlighting this error. We have now gone through all references systematically to ensure essential information is included. For example, see Refs 1,30, and 41.

2. The term 'mannotation' is still used in the Supplementary Materials (e.g. Fig S1, Dataset 2); I would suggest just adding a note describing your short-hand notation somewhere in the manuscript (just before the legends for Supplementary data, for example).

We have added brief explanatory notes in the captions where “mannotation” is used. (lines 1257 & 1382)

3. Fig. S2B - match/mismatch colours mixed up? (I think match should be teal and mismatch - red?)

The legend was previously mismatched with the labels. This has been corrected in the new uploaded figure.

4. Line 162-163: Rv1430 is in UniProt (EC 3.1.1.-) and has been present in Uniprot since version 45 of the gene record: <https://www.uniprot.org/uniprot/L7N697>. I presume you had conducted your literature analysis before the UniProt entry was updated to include the EC code, so maybe you can add the dates when the data was retrieved from UniProt and other databases you used in the Materials and Methods section?

The reviewer's presumption is correct; we had stated the date of data retrieval in the caption of Table 1, but we agree it should instead be stated centrally in the Methods. We have now added it to the Methods section as well, for clarity (Lines 696-700)

5. Supplementary text, p. 9, first paragraph. I believe that an unrelated fragment of text was copy-pasted into the second sentence of the paragraph ("Many mutations that altered bacterial clearance...")

We thank the reviewer for catching this accidental insertion. We have now removed the spurious fragment.

6. Supplementary text, p. 12, final paragraph. It should be Rv1191, not Rv1191c. Could you also add a short explanation why you believe it should be classified as a cathepsin (what protein did you transfer this annotation from)?

We have removed this speculation in the revised submission.

Reviewer #3 (Comments for the Author):

In this manuscript, Modlin et al., attempt to tackle the problem of assigning functions to ~1700 hypothetical and/or under-annotated genes in the Mycobacterium tuberculosis H37Rv (Mtb) genome. Rapid and accurate annotation of microbial genomes is indeed a very critical and under appreciated part of microbial ecophysiology. This step is especially crucial for pathogenic organisms such as Mtb where accurate functional annotation of these hypothetical proteins could unravel mechanisms which could act as drug targets.

The authors employed a two-pronged strategy to define a set of these unannotated or under-annotated genes and to then provide possible functions for many of these genes. First, they undertook a large-scale manual curation of literature to assign functions (including EC numbers for enzymatic functions) to ~575 genes. These annotations were compared across the known annotation resources for Mtb such as UniProt, the Mtb Network Portal, PATRIC, and RefSeq. The second approach used by the authors involved a pipeline for structural modelling of proteins to infer function based on shared 3D structure. Combining these two approaches the authors were able to provide annotations for 623 such under-annotated proteins.

The manuscript is generally well written (given the fact that it has already been reviewed before) and easy to follow. The methods employed are sound and extensive, and the results are described and discussed adequately.

Homology based protocols are limited by the lack of sufficient sequence similarity and can be complemented by context based methods. This is one aspect that the authors have not discussed in detail.

We thank the reviewer for this review and feedback. As the discussion and overall length of this manuscript is already quite long, we do not feel that further explication of this consideration is necessary to support the claims in this manuscript. Especially since we do mention this consideration, and use context to further inform hypotheses for Rv2706c.

The central idea of expanding the functional repertoire of Mtb is not novel and even structure based alignment has been attempted before (PMID: 31113361, PMID: 33076816). As such, the combined approach of extensive literature survey and the large-scale structural modelling is unique. How do the novel predictions of function in this manuscript match up with other similar predictions?

We intentionally opted not to systematically compare previous structure-based alignment on Mtb because those works used modelling software with confidence metrics entirely uncalibrated to the likelihood that a structural homolog would be correct. This stands in contrast to our implementation of I-TASSER and downstream analysis of its output: We employ the previously determined benchmarks for how C-score and TM-score provided relate to probability of genuine functional and structural family matches (lines 176-177; 541-543; 608-640; 1385-1388).

As such, we believe such a systematic comparison adds little to the knowledge presented in this manuscript and distracts from the core messages in an already data-heavy report. Therefore, we have not included additional content systematically comparing our results to the study mentioned by the reviewer.

October 4, 2021

Dr. Faramarz Valafar
San Diego State University
Laboratory for Pathogenesis of Clinical Drug Resistance and Persistence
6367 Alvarado Court
Suit 206
San Diego, California 92120

Re: mSystems00673-21R1 (Structure-aware *M. tuberculosis* functional annotation uncloaks resistance, metabolic, and virulence genes)

Dear Dr. Faramarz Valafar:

Your manuscript has been accepted, and I am forwarding it to the ASM Journals Department for publication. For your reference, ASM Journals' address is given below. Before it can be scheduled for publication, your manuscript will be checked by the mSystems senior production editor, Ellie Ghatineh, to make sure that all elements meet the technical requirements for publication. She will contact you if anything needs to be revised before copyediting and production can begin. Otherwise, you will be notified when your proofs are ready to be viewed.

As an open-access publication, mSystems receives no financial support from paid subscriptions and depends on authors' prompt payment of publication fees as soon as their articles are accepted. =

Publication Fees:

We recognize that the video files can become quite large, and so to avoid quality loss ASM suggests sending the video file via <https://www.wetransfer.com/>. When you have a final version of the video and the still ready to share, please send it to Ellie Ghatineh at eghatineh@asmusa.org.

Sincerely,

John McGrath
Editor, mSystems

Journals Department
Dataset 1: Accept
Dataset 2: Accept
Dataset 3: Accept
Figure S1: Accept
Supplemental Material 8: Accept
Figure S5: Accept
Figure S4: Accept
Supplemental Material: Accept
Figure S3: Accept
Figure S2: Accept